# Geometry of population activity in spiking networks with low-rank structure

**Ljubica Cimeša**[ID]**, Lazar Ciric, Srdjan Ostojic**[ID]*

Laboratoire de Neurosciences Cognitives Computationnelles, Département d'Études Cognitives, École Normale Supérieure, INSERM U960, PSL University, Paris, France

* srdjan.ostojic@ens.fr

**Data Availability Statement:** Code available at: https://github.com/LjubicaCimesa/Spiking-low-rank-networks.

**Funding:** The project was supported by the CRCNS project PIND (ANR-19-NEUC-0001-01 to SO), the program "Ecoles Universitaires de Recherche"

## Abstract

Recurrent network models are instrumental in investigating how behaviorally-relevant computations emerge from collective neural dynamics. A recently developed class of models based on low-rank connectivity provides an analytically tractable framework for understanding of how connectivity structure determines the geometry of low-dimensional dynamics and the ensuing computations. Such models however lack some fundamental biological constraints, and in particular represent individual neurons in terms of abstract units that communicate through continuous firing rates rather than discrete action potentials. Here we examine how far the theoretical insights obtained from low-rank rate networks transfer to more biologically plausible networks of spiking neurons. Adding a low-rank structure on top of random excitatory-inhibitory connectivity, we systematically compare the geometry of activity in networks of integrate-and-fire neurons to rate networks with statistically equivalent low-rank connectivity. We show that the mean-field predictions of rate networks allow us to identify low-dimensional dynamics at constant population-average activity in spiking networks, as well as novel non-linear regimes of activity such as out-of-phase oscillations and slow manifolds. We finally exploit these results to directly build spiking networks that perform nonlinear computations.

## Author summary

Behaviorally relevant information processing is believed to emerge from interactions among neurons forming networks in the brain, and computational modeling is an important approach for understanding this process. Models of neuronal networks have been developed at different levels of detail, with typically a trade off between analytic tractability and biological realism. The relation between network connectivity, dynamics and computations is best understood in abstract models where individual neurons are represented as simplified units with continuous firing activity. Here we examine how far the results obtained in an analytically-tractable class of rate models extend to more biologically realistic spiking networks where neurons interact through discrete action potentials. Our results show that abstract rate models provide accurate predictions for the collective dynamics and the resulting computations in more biologically faithful spiking networks.

launched by the French Government and implemented by the ANR, with the reference ANR-17-EURE-0017 (to LC and SO). The funders had no role in study design, data collection and analysis, decision to publish, or preparation of the manuscript.

**Competing interests:** The authors have declared that no competing interests exist.

## Introduction

Recurrent network models are an essential tool for understanding how the collective dynamics of activity in the brain give rise to computations that underlie behavior. Network models at different levels of biological detail are typically used to describe different phenomena [1, 2], but integrating findings across scales of abstraction remains challenging. Networks of excitatory and inhibitory spiking neurons [3] are a popular class of models which incorporate the key biological fact that neurons interact through discrete action potentials, a.k.a. spikes. Balanced excitatatory-inhibitory spiking networks in particular naturally lead to asynchronous irregular activity [4–12] that captures some of the main features of the spontaneous neural firing in vivo [13–19]. Beyond spontaneous activity, how rich behavioral computations are implemented in spiking networks has been an open issue [20–22]. This question has so far been more easily tackled in more abstract models such as recurrent neural networks (RNNs) [23–25], where individual units are represented in terms of continuous firing rates rather than discrete spikes. A particularly fruitful approach has been to interpret the emerging computations in terms of the geometry of dynamics in the state space of joint activity of all neurons [26–29], as commonly done with experimental data [30–35]. In particular, in a large class of rate networks in which the connectivity contains a low-rank structure [36–48], the geometry of activity and the resulting computations can be analytically predicted from the structure of connectivity [49–52]. A comparable mechanistic picture has so far been missing in spiking networks.

A key question is therefore to which extent mechanistic insights from RNNs extend to more biologically plausible spiking models. In this regard, a central underlying issue is exactly how spiking models are related to abstract rate networks [53, 54]. This question has been addressed in various specific cases [55–61], but a systematic mathematical reduction of arbitrary spiking networks to rate models has been elusive. One common heuristic has been to interpret each unit in a rate network as an average over a sub-population of spiking neurons [4, 62–69]. A possible alternative is instead to approximate each individual spiking neuron by a Poisson rate unit [70], and therefore hypothesize that a full spiking network can be directly mapped onto a rate network with identical connectivity [71–76]. If this hypothesis is correct, the analytic predictions for the geometry of activity in rate networks should directly translate to spiking networks with a low-rank structure in connectivity. This implies that the geometry of activity and range of dynamics in spiking networks may be much broader than apparent on the level of population-averaged spike trains.

To test this hypothesis, we consider a classical spiking network model consisting of excitatory-inhibitory integrate-and-fire neurons [7], and add low-rank structure on top of the underlying random, sparse connectivity. Varying the statistics of the low-rank structure, we systematically compare the geometry of activity and dynamical regimes in the spiking model with predictions of networks of rate units with statistically identical low-rank connectivity. We find that rate networks predict well the structure of activity in the spiking network even outside the asynchronous irregular regime, as long as spike-times are averaged over timescales longer than the synaptic and membrane time constants to estimate instantaneous firing rates. In particular, the predictions of the rate model allow us to identify low-dimensional dynamics at constant population-average activity in spiking networks, as well as novel non-linear regimes of activity such as out-of-phase oscillations and slow manifolds. We finally show that these results can be exploited to directly build spiking networks that perform nonlinear computations based on principles identified in rate networks.

## Results

### Geometry of the activity in the state space

We consider recurrent networks of $N$ neurons, modeled either as rate units or leaky integrate-and-fire (LIF) neurons (Fig 1A, see Methods for details). We quantify the activity of each neuron $i$ in terms of its time-dependent firing rate $r_i(t)$. In rate networks, each unit is described by the dynamics of its activation $x_i(t)$, an abstract variable usually interpreted as the total input or membrane potential [77], that obeys

$$\tau \dot{x}_i(t) = -x_i(t) + \sum_{j=1}^{N} P_{ij}\,\phi(x_j) + I_i u(t). \tag{1}$$

Here $P_{ij}$ is the recurrent connectivity weight from unit $j$ to unit $i$, $u(t)$ is the input amplitude shared by all units, $I_i$ is the weight of the external input on unit $i$, and the firing rate is obtained as $r_i(t) = \phi(x_i(t))$, where $\phi(x)$ is the single-unit current-to-rate function that we here choose to be a positive sigmoid $\phi(x) = 1 + \tanh(x - x_{off})$. In the LIF network, single-unit firing rates are instead estimated from a running average over spike times, computed using an exponential filter with timescale $\tau_f$ (Fig 1C, Methods Eq (18)).

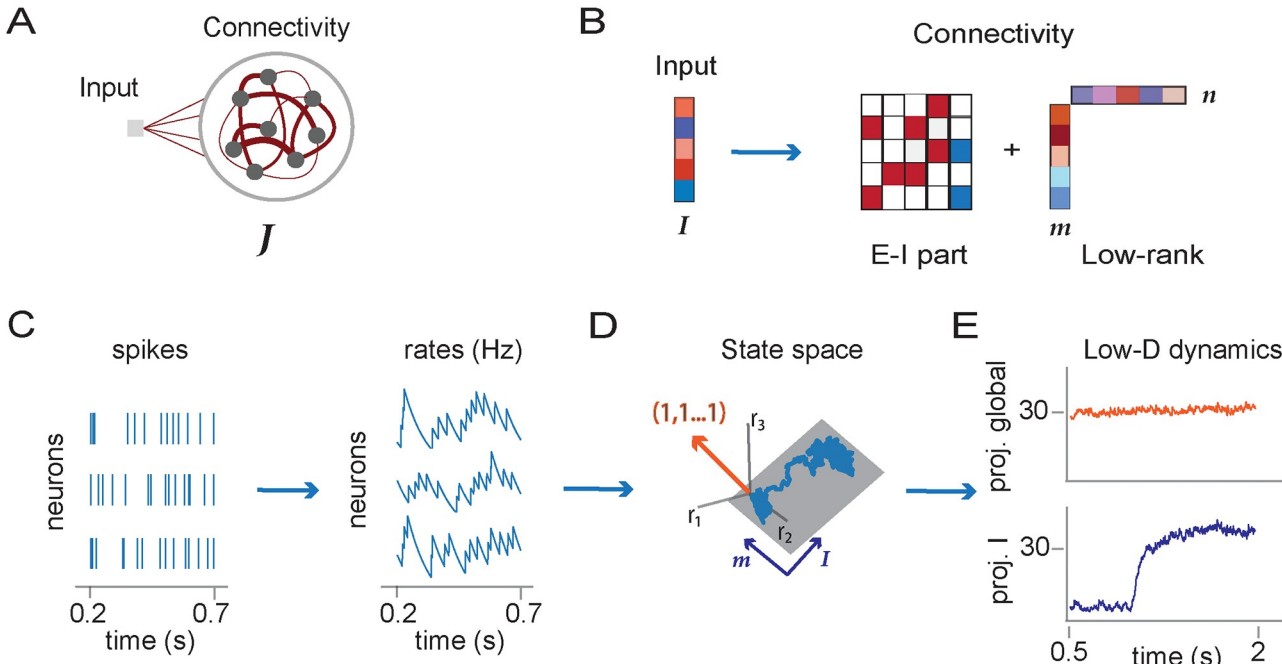

**Fig 1. Low-rank connectivity and state space dynamics.** A: Illustration of recurrent neural network architecture, consisting of inputs and recurrent connectivity. B: Representation of inputs and connectivity in terms of vectors. The input weights form an input vector $\boldsymbol{I}$. In spiking networks, the recurrent connectivity is composed of a sparse excitatory-inhibitory part (zero entries in white, excitatory connections in red, inhibitory in blue) and a low-rank structure defined by pairs of connectivity vectors $\boldsymbol{m}$ and $\boldsymbol{n}$. The illustration shows a unit-rank example ($R = 1$). C: Left: Spike times of three neurons in the spiking network. Right: dynamics of instantaneous firing rates computed from spikes using an exponential filter with timescale $\tau_f =$ 100ms. D: Three-dimensional illustration of low-dimensional dynamics in the activity state space where each axis represents the firing rate of one neuron. In a unit-rank network, the activity is expected to be confined to a two-dimensional plane spanned by the vectors, $\boldsymbol{m}$ and $\boldsymbol{I}$. We refer to the direction $(\boldsymbol{1}, \boldsymbol{1}, \ldots \boldsymbol{1})$ as the global axis (orange). E: Projections of activity on two axes: (top) global axis corresponding to the population-averaged firing rate; (bottom) axis defined by the input vector $\boldsymbol{I}$.

Following a common approach for analyzing neural data [32, 78], we represent the collective activity at any time point as a vector $\boldsymbol{r}(t) = \{r_i\}_{i = 1\dots N}$ in the *activity state space* where each axis corresponds to the firing rate $r_i$ of one neuron (Fig 1D). We then examine the geometry of the dynamical trajectories by projecting at each time point the activity vector $\boldsymbol{r}(t)$ along different directions in that space. Each direction is specified by a vector $\boldsymbol{w} = \{w_i\}_{i = 1\dots N}$ in state space, so that projecting onto it is equivalent to assigning to every neuron a weight $w_i$ and computing a weighted average of the activity (Methods Eq (20)).

Analyses of experimental data and works on rate networks commonly examine how collective activity changes along arbitrary directions in state space, where the weight $w_i$ of each neuron is chosen independently. In contrast, studies of spiking networks have often focused on firing rates averaged over the whole network or over specific sub-populations [4, 63–68]. Taking a population average over the whole network is equivalent to projecting activity along the direction $(\boldsymbol{1}, \boldsymbol{1}, \dots, \boldsymbol{1})$, which we call the *global axis* [79]. Similarly, vectors with unit entries on a specific subset of neurons and zeros elsewhere define directions in state space that represent firing rates averaged over specific sub-populations. The goal of the present study is to instead examine how inputs and connectivity in spiking networks shape activity along arbitrary directions of state space, and in particular directions orthogonal to the global axis which correspond to changes in collective activity that modify the pattern of activity but keep the population-averaged activity constant. To this end, we compare the geometry of activity in spiking networks with rate networks that share an identical part of the connectivity $\boldsymbol{P}$.

We specifically focus on rate networks with a low-rank connectivity matrix parametrized as

$$P_{ij} = \frac{1}{N}\sum_{r=1}^{R} m_i^{(r)} n_j^{(r)} \tag{2}$$

where $\boldsymbol{m}^{(r)} = \{m_i^{(r)}\}_{i=1\dots N}$ and $\boldsymbol{n}^{(r)} = \{n_i^{(r)}\}_{i=1\dots N}$ for $r = 1, \dots, R$ are *connectivity vectors* (Fig 1B). Rate networks with such a connectivity are analytically tractable, in the sense that the geometry of dynamics in state space can be directly predicted from the arrangement of connectivity vectors and inputs, as summarized below.

Our key hypothesis is that, for a spiking network in the asynchronous irregular state, each neuron can be directly mapped onto a unit in a rate network with statistically identical connectivity. To test this hypothesis, we start from an LIF network with sparse excitatory-inhibitory connectivity $\boldsymbol{J}^{EI}$, and choose parameters in the inhibition-dominated regime that leads to asynchronous irregular activity [7] (Methods). We add to this random component a low-rank part $\boldsymbol{P}$ given by Eq (2). We then compare the geometry of the resulting spiking activity to the predictions of a rate model with low-rank connectivity $\boldsymbol{P}$.

### Geometry of responses to external inputs

We start by examining the geometry of transient dynamics in response to external inputs. We first summarize the predictions of low-rank rate models developed in previous studies [49]. We then examine whether these predictions hold in networks of integrate-and-fire models with statistically identical low-rank components.

Each neuron $i$ receives a step input $u(t)$ (Fig 2B) multiplied by a weight $I_i$. The set of feed-forward weights $I_i$ over neurons form an *input vector* $\boldsymbol{I} = \{I_i\}_{i = 1\dots N}$. This input vector, as well as the connectivity vectors $\boldsymbol{m}^{(r)}$ and $\boldsymbol{n}^{(r)}$ introduced in Eq 2, each define a specific direction in state space (Fig 2A). Previous work [49, 51] has shown that in rate networks with low-rank connectivity, the dynamics of the activations $\boldsymbol{x}(t) = \{x_i(t)\}$ in response to an input are confined to a subspace of state space spanned by the input vector $\boldsymbol{I}$ and connectivity vectors $\boldsymbol{m}^{(r)}$ for $r = 1\dots R$. Focusing on a unit rank network ($R = 1$), this implies that the activation $x_i(t)$ of unit

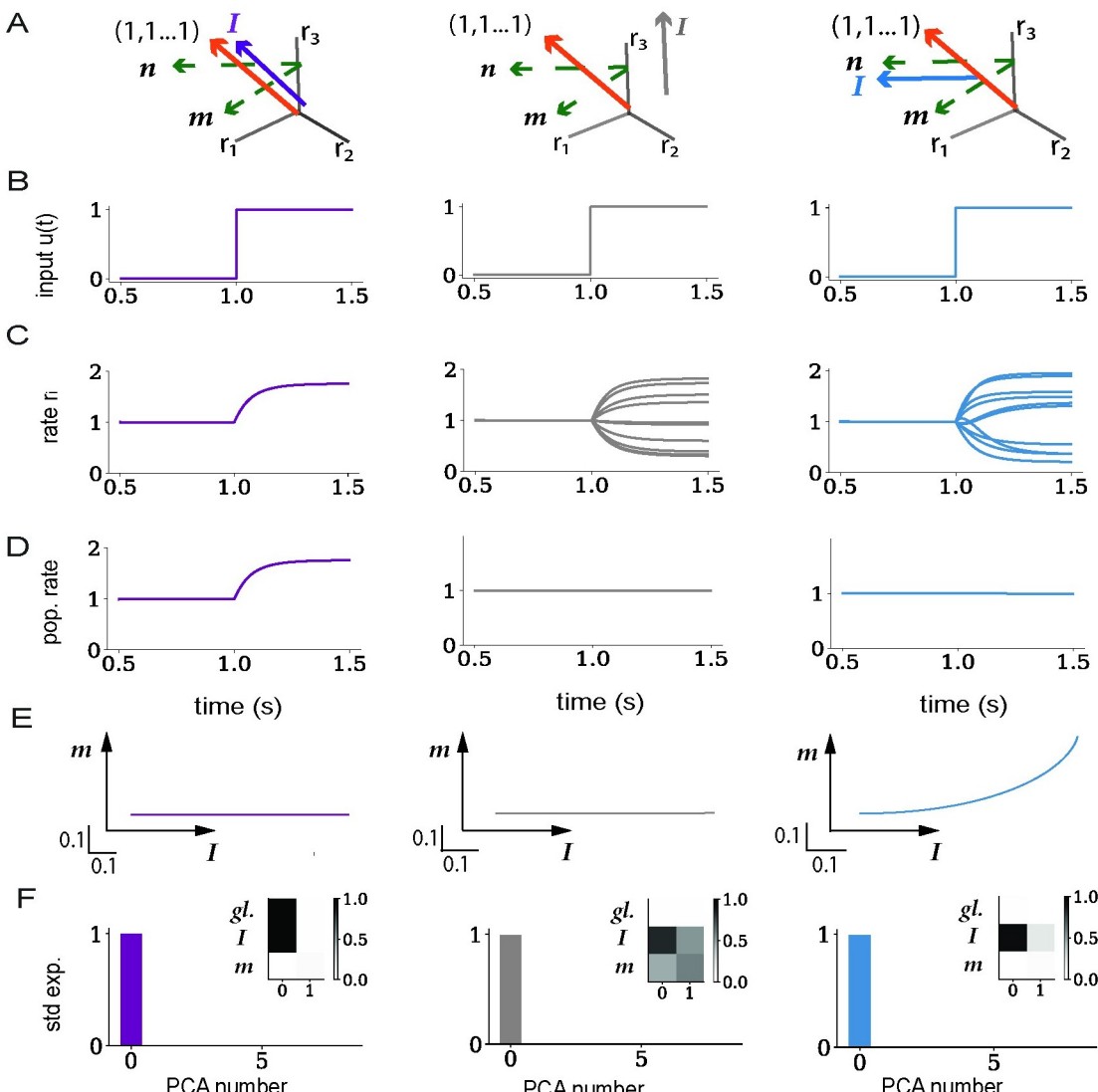

**Fig 2. Low-dimensional dynamics generated by external inputs in rate networks with low-rank connectivity.** A: Illustration of the geometry in the activity state-space. The input vector $I$, the connectivity vectors $m$ and $n$ (green), and the global axis $(1, 1, \ldots, 1)$ (orange) define a set of directions and a subspace within which the low-dimensional dynamics unfold. The overlaps of the vectors $I$ and $m$ with the global axis predict whether inputs give rise to a change in the population-averaged activity. The overlap of $I$ with $n$ instead determines whether an input engages recurrent activity along the direction $m$. The three columns display three different arrangements of the input vector (depicted in a different color in each column). Left: $I$ aligned with the global axis; middle: $I$ orthogonal to both $n$ and the global axis; right: $I$ aligned with $n$, but $I$ and $m$ orthogonal to the global axis. B: Input vector is multiplied by a scalar $u(t)$ which is a step function from $t = 1$s. C: Individual firing rates $r_i(t)$ for a subset of 10 neurons in each network. D: Population firing rate, averaged over all neurons in the network. E: Projections of the firing rate trajectory $r(t)$ onto the $(I,m)$ plane. F: PCA analysis of the firing rate dynamics $r(t)$. Variance explained by each of the first 8 PCs. Inserts: Projections of the first two PCs onto the global axis, the input vector $I$ and the connectivity vector $m$. The connectivity vectors $m$ and $n$ have a zero mean and unit standard deviation, and are orthogonal to each other. Vectors $n$ and $I$ are orthogonal except in blue where the overlap is $\sigma_{nI} = 1$. Vectors $I$ in gray and blue have a zero mean and unit standard deviation, while vector $I$ in purple is along the global axis. Network parameters are given in Table 1.

$i$ in the rate network can be expressed as

$$x_i(t) = \kappa(t)\, m_i + \nu(t) I_i \tag{3}$$

where $\kappa(t)$ and $\nu(t)$ are two scalar variables (Methods). The variable $\nu(t)$ represents feed-forward activity propagated along the direction $\boldsymbol{I}$, while $\kappa(t)$ quantifies activity that recurrent dynamics generate along the direction $\boldsymbol{m}$. An input along the direction $\boldsymbol{I}$ will generate a non-zero recurrent response $\kappa(t)$ only if $\boldsymbol{I}$ has a non-zero overlap with the vector $\boldsymbol{n}$, i.e., if the scalar product $\boldsymbol{n}^T \boldsymbol{I}$ is non-zero [49].

Additional analyses show that the low-dimensional geometry described by Eq (3) at the level of activations $\boldsymbol{x}(t)$ is largely preserved when applying the non-linear function $\phi(x)$ to obtain rates (Methods). More specifically, the projection of the firing rates $\boldsymbol{r}(t) = \{r_i(t)\}$ on an arbitrary axis $\boldsymbol{w}$ is determined by the projection of $\boldsymbol{x}(t)$ on that same axis if the response was weak and therefore approximately linear, or if the entries of $w_i$ of $\boldsymbol{w}$ follow a Gaussian distribution (Eq (34)), which we assume throughout this study. In low-rank rate networks, the dynamics of $\boldsymbol{r}(t)$ in response to an input therefore dominantly lie in the subspace spanned by the input and connectivity vectors $\boldsymbol{I}$ and $\boldsymbol{m}$ (Fig 2E), with the non-linearity generating a potential additional component along the global axis (Methods Eq (34)). These theoretical predictions were confirmed by a PCA analysis of simulated trajectories of firing rates $\boldsymbol{r}(t)$ (Fig 2F).

Since the population-averaged firing rate is obtained by projecting $\boldsymbol{r}(t)$ on the global axis ($\boldsymbol{1}$, $\boldsymbol{1}, \ldots, \boldsymbol{1}$), the analysis of low-rank rate models predicts that a given input induces a strong change of population-averaged firing rates when the mean of inputs weights over neurons, $\langle I \rangle = \frac{1}{N} \sum_{i=1}^{N} I_i$, is non-zero (Fig 2D, left panel), or if both the average $\langle m \rangle$ of elements of $\boldsymbol{m}$ and the overlap $\boldsymbol{n}^T \boldsymbol{I}$ are non-zero. Conversely, if $\langle I \rangle = 0$ and $\langle m \rangle = 0$, inputs evoke changes in single-unit firing rate that essentially average-out on the population-average level (Fig 2D, middle and right panel), but instead explore the plane $\boldsymbol{I} - \boldsymbol{m}$ that is orthogonal to the global axis in state space (Fig 2E, middle and right panel).

We compared these predictions of low-rank rate models to the geometry of activity in spiking networks where a low-rank structure $\boldsymbol{P}$ was added on top of sparse excitatory-inhibitory connectivity $\boldsymbol{J}^{EI}$ (Fig 3). Note that, because of the $1/N$ scaling in Eq (2), the magnitude of elements of $\boldsymbol{P}$ was much smaller than the non-zero elements of $\boldsymbol{J}^{EI}$ that were independent of $N$ [7]. The excitatory-inhibitory part of the connectivity therefore controlled the firing regime of the network. Starting from a network in the inhibition-dominated asynchronous irregular state [7], as expected inputs evoked a change in population-averaged firing rates if the mean of the input vector $\langle I \rangle$ was non-zero (Fig 3D, left panel). Input vectors of zero mean instead elicited patterns of responses across neurons that did not modify the population-averaged firing rate (Fig 3D, middle and right panel), but explored directions in state space orthogonal to the global axis. These directions were accurately predicted by low-rank rate models: input vectors $\boldsymbol{I}$ orthogonal to the vector $\boldsymbol{n}$ led to responses only along the direction $\boldsymbol{I}$ (Fig 3E, left and middle panel), while inputs that overlapped with $\boldsymbol{n}$ led to responses in the $\boldsymbol{I} - \boldsymbol{m}$ plane (Fig 3E, right panel). A PCA analysis confirmed that these low-dimensional projections explained the dominant part of variance in the full trajectories (Fig 3F).

Because the low-rank structure $\boldsymbol{P}$ is generated using Gaussian connectivity vectors, the full connectivity matrix obtained by superposing it with the random component $\boldsymbol{J}^{EI}$ does not obey Dale's law and is not sparse. The entries of $\boldsymbol{P}$ are however much weaker (standard deviation $\sigma_n * \sigma_m / N$) than the non-zero elements of $\boldsymbol{J}^{EI}$. Setting to zero the entries of $\boldsymbol{P}$ for which $\boldsymbol{J}^{EI}$ is zero was therefore sufficient to enforce Dale's law in the full connectivity matrix. The resulting sparsified matrix $\boldsymbol{P}$ is not low-rank anymore, but for rate networks the predictions of low-rank theory still hold up to high values of sparsity [80]. We verified that the results in the spiking

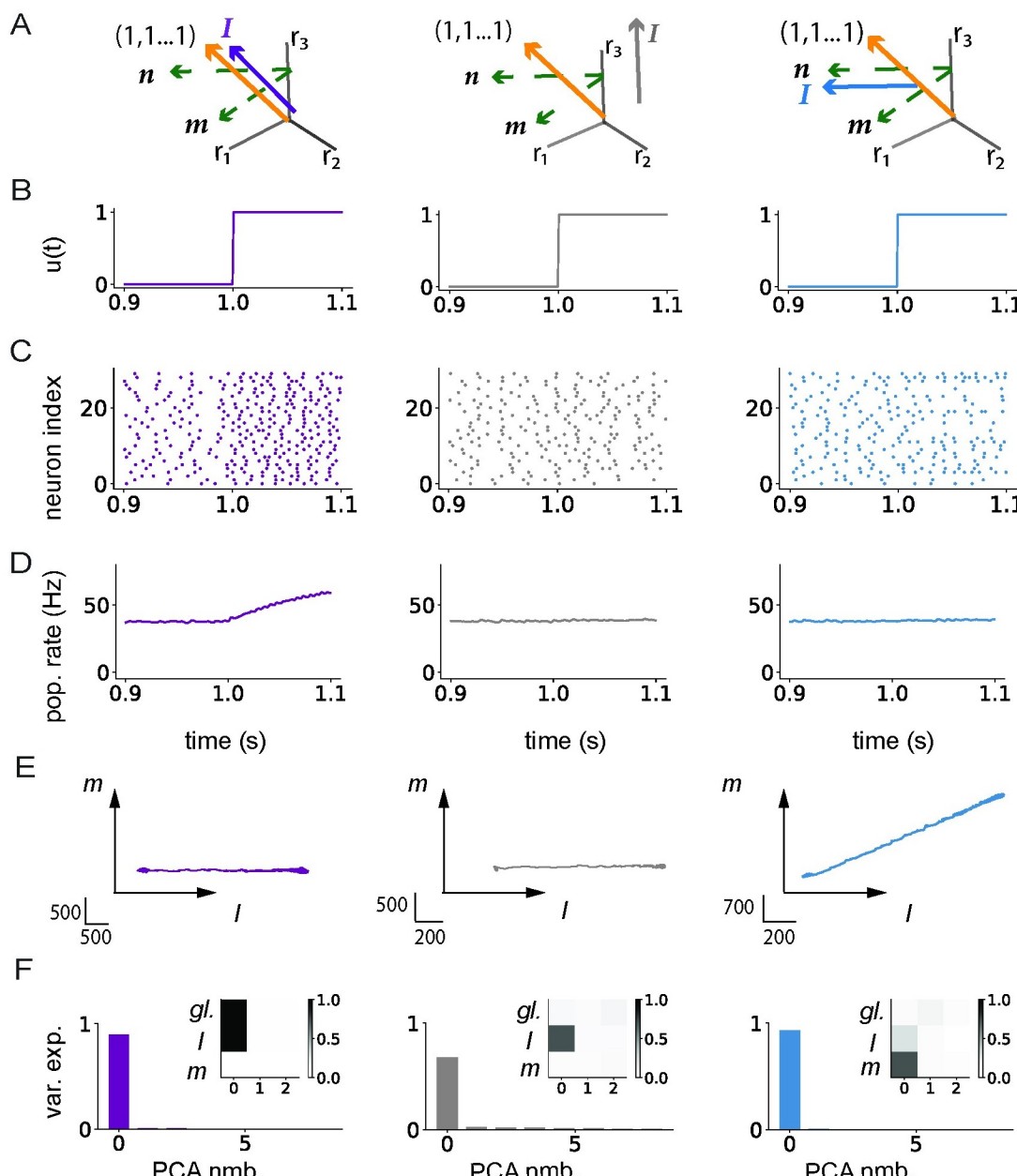

**Fig 3. Low-dimensional dynamics generated by external inputs in spiking networks with low-rank structure.** A: Illustration of the geometry of input (varying color) and connectivity vectors (green) with respect to the global axis (orange). Left: input vector $I$ along the global axis; middle: input vector $I$ orthogonal to $n$; right: input vector $I$ along the vector $n$. B: Input vector is multiplied by a scalar $u(t)$ which is a step function from $t = 1s$. C: Raster plot showing action potentials for a subset of 30 neurons out of $N = 12500$ in each network. D: Population firing rate obtained by averaging instantaneous firing rates of all neurons. E: Projections of the firing rate trajectory $r(t)$ onto the $(I, m)$ plane. F: PCA analysis of firing rate dynamics $r(t)$. Variance explained by each of the first 8 PCs. Inserts: Projections of the first 3 PCs onto the global axis (first row), and vectors $I$ and $m$. The connectivity vectors $m$ and $n$ have a zero mean and unit standard deviation, and are orthogonal to each other. Vectors $n$ and $I$ are orthogonal except in blue where the overlap is $n^T I/N = 0.4\text{mV}^2$. Vectors $I$ in gray and blue have a zero mean and unit standard deviation, while vector $I$ in purple is along the global axis. All analyses were performed on instantaneous firing rates computed using a filter timescale of $\tau_f = 100\text{ms}$. Network parameters are given in Table 2.

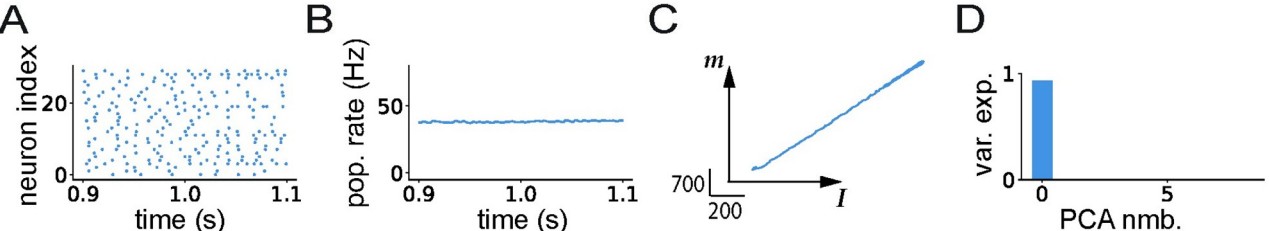

**Fig 4. Low-dimensional dynamics generated by external inputs in spiking network in which the full connectivity is sparse and satisfies Dale's law.**
The connectivity consisted of the superposition of the random term $J^{EI}$ and a sparsified unit-rank part $P$, in which we set to zero entries for which $J_{ij}^{EI} = 0$. The input vector $I$ is along the vector $n$. A: Raster plot showing action potential for a subset of 30 neurons. B: Population firing rate obtained by averaging instantaneous firing rates of all neurons. C: Projection of the firing rate trajectory $r(t)$ onto the $(I, m)$ plane. D: PCA analysis of firing rate dynamics $r(t)$. Variance explained by each of the first 8 PCs parameters in Table 2.

network were unchanged when including sparsity in the low-rank term and Dale's law in the full connectivity (Fig 4).

Altogether, the predictions of low-rank rate models were fully borne out when treating each individual spiking neuron as a rate unit.

## Responses in spiking networks outside of the irregular asynchronous regime

Our initial hypothesis was that low-rank rate networks predict well the geometry of responses of spiking networks in the asynchronous irregular regime, where individual neurons can be approximated as independent Poisson processes [7]. We next asked to which extent the predictions hold outside of this regime, when the spiking activity is either not asynchronous, i.e. exhibits some degree of synchronization and oscillations [7], or is regular rather than irregular. Following Brunel 2000, we set the network to operate in a specific regime by varying the strength of the inhibition $g$ in the random part of the connectivity, the external input $\mu_{ext}$ and the synaptic delay $\tau_{del}$ (Methods). We then examined how much the underlying regime influences the low-dimensional dynamics in response to external inputs in networks with a unit-rank structure. For this, we repeated the PCA analysis in spiking networks with zero-mean input and connectivity vectors, and $I$ overlapping with $n$ as in the right column of Fig 2.

We first considered a network of integrate-and-fire neurons that operates in the synchronous irregular (SI) regime [7] in which individual neurons fire irregularly (Fig 5A, top), but are sparsely synchronised, leading to oscillations in the population rate (Fig 5A, bottom). The frequency of these oscillations is set by the synaptic delay $\tau_{del}$, and is therefore high for physiologically realistic values of $\tau_{del}$ [7]. These oscillations can therefore only be observed in the firing rates when the filter timescale $\tau_f$ used for averaging over spikes is comparable to the delays, i.e. of the order of milliseconds (Fig 5A, blue). Longer filter timescales instead totally average-out the oscillatory dynamics (Fig 5A, orange). We therefore found that the dimensionality and geometry of the responses in state space depend on the filter timescale used to determine single-unit firing rates (Fig 5B). Performing a PCA analysis on firing rate trajectories $r(t)$ obtained with a filter timescale of 1ms indicated that the activity was high-dimensional. Indeed, the explained variance was distributed along many principal components (Fig 5B, top), with the first PC capturing population-level oscillations along the global axis (insert in Fig 5B, top panel), while strong fluctuations were present in other directions (Fig 5C). In contrast, for a filter timescale of 100ms the first PC explained a much larger fraction of variance (Fig 5B, bottom), and corresponds instead to activity along a combination between the connectivity vector

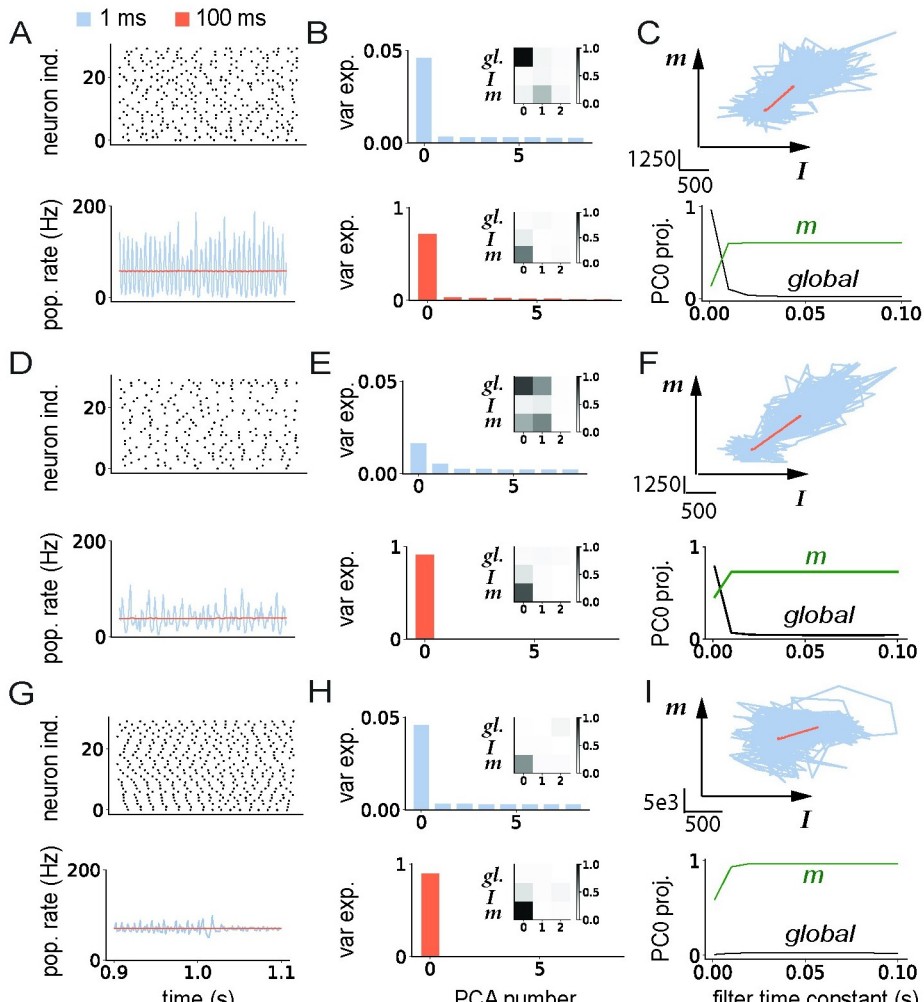

**Fig 5. Influence of firing regime and filter timescales on low-dimensional dynamics in spiking networks.** A-C: Synchronous irregular (SI) regime. A: Top: Raster plot showing action potentials for a subset of 30 neurons in the network. Bottom: population-averaged firing rate computed using filter time constants of 1ms (blue) and 100ms (orange). B: PCA analysis of trajectories of instantaneous firing rates computed from spike trains using two different filter time constants (top: 1ms, bottom: 100ms). Main panels: variance explained by each of the first 8 PCs; inserts: projections of the first 3 principal components on the global vector, $I$ and $m$. C: Top: Projections of the firing rate trajectories on the plane defined by vectors $m$ and $I$. Bottom: Projection of the first principal component on the global axis (black) and on the vector $m$ (green) as a function of the filter time constant. D-F: Similar to A-C, for the network in asynchronous irregular regime shown in the right column of Fig 3. G-I Similar to A-C, for a network without the background E-I connectivity. The firing regime was controlled by varying the inhibition strength in the random EI connectivity, the baseline input and synaptic delays (see Table 2). The unit-rank connectivity structure was identical to Fig 3 right column, with zero-mean input and connectivity vectors. At time $t = 1$s, a step input was given along the input vector $I$ that was aligned with $n$. Network parameters are given in Table 2.

$m$ and the input vector $I$ (insert in Fig 5B, bottom), as predicted by the rate model (insert in Fig 2F, blue). In between these two extremes, progressively increasing the filter timescale (Fig 5C, bottom) shows that for timescales below 10ms, the geometry of activity is dominated by fluctuations along the global axis, while for longer timescales the dynamics are lower-dimensional and reside dominantly in the ($m$, $I$) plane as expected from the rate network (Fig 2E and 2F, blue).

Given the strong influence of the filter timescale on the results, we repeated the same analysis in the asynchronous irregular (AI) regime, which in Fig 3 was investigated only using a

long timescale of 100ms. We found that the results of the PCA were similar to the SI regime: fluctuations along the global axis dominated at timescales below 10ms, and low-dimensional dynamics predicted by the rate model emerged at longer timescales (Fig 5D–5F). The main difference between the AI and SI regimes is that the global fluctuations at short timescales are weaker in the AI regime (with an amplitude that decays as the network size is increased), and do not show the periodic structure found in the SI regime (Fig 5D).

We then examined the role of irregular activity, by turning to networks in which the connectivity consisted only of a low-rank structure without the random E-I part. As noted above, on the level of individual synapses this removed the dominant part of the connectivity, as the magnitude of non-zero terms in $J^{EI}$ was much larger than the magnitude of terms in $P$. In such networks, individual neurons fired almost periodically, in contrast to Poisson-like activity in the AI regime. The action potentials of the different neurons were however highly asynchronous (Fig 5G top), and the fluctuations in the population activity were weak even for filter timescales of 1ms (Fig 5G bottom). Similarly to SI and AI regime, the dynamics in this network became low-dimensional for long filter timescales (Fig 5H and 5I), but the projection along $m$ was higher for all filter timescales, and saturated above 10ms (Fig 5I bottom).

In summary, our analyses indicate that the predictions of the rate networks for the geometry of responses hold in different activity regimes in the spiking network if the single neuron firing rates are determined by averaging action potentials on a timescale longer than the synaptic delays. At shorter timescales, the activity is dominated by spiking synchronization that leads to prominent fluctuations along the global axis which corresponds to the population-averaged firing rate.

## Nonlinear autonomous activity in networks with unit-rank structure

In previous sections we studied the geometry of dynamics in response to external inputs. We next turned to autonomous dynamics generated by the recurrent connectivity in the absence of inputs. As before, our goal was to determine whether the dynamics in a spiking network with low-rank connectivity are well predicted by a rate network with an analogous low-rank structure in the connectivity. We first summarize the results for rate networks developed in earlier studies, and then compare dynamics in spiking networks with these predictions.

In a rate network with unit rank structure, in absence of time-varying external inputs, the low-dimensional dynamics in Eq (3) are described only by the recurrent variable $\kappa(t)$. The temporal evolution of $\kappa(t)$ obeys (Methods):

$$\frac{d\kappa}{dt} = -\kappa + \frac{1}{N}\sum_{i=1}^{N}\phi(x_i)\,n_i. \tag{4}$$

The steady state state value of $\kappa$ therefore satisfies:

$$\kappa = \sum_{i=1}^{N} n_i\,\phi(x_i)/N. \tag{5}$$

where $x_i$ is the steady state value of $x_i(t)$.

Assuming as previously a Gaussian distribution of the entries $(m_i, n_i)$ of the connectivity vectors, and using a mean-field analysis in the large $N$ limit, Eq 5 can be further expressed as (Methods Eq (40))

$$\kappa = \langle n \rangle \langle \phi(\mu, \Delta) \rangle + \sigma_{mn} \kappa \langle \phi'(\mu, \Delta) \rangle \tag{6}$$

where $\langle m \rangle$, $\langle n \rangle$ and $\sigma_{mn}$ are the mean values and covariance of connectivity vectors $\boldsymbol{m}$ and $\boldsymbol{n}$, while $\langle \phi(\mu, \Delta) \rangle$ and $\langle \phi'(\mu, \Delta) \rangle$ are the mean firing rate and mean gain obtained by averaging

the transfer function and its derivative over a Gaussian distribution of mean $\mu = \langle m \rangle \kappa$ and variance $\Delta = \sigma_m^2 \kappa^2$ (Methods Eq (40), Methods Eq (51)). Eq (6) provides a self-consistent equation for the steady state value of $\kappa$, which enters implicitly in the r.h.s. through $\mu$ and $\Delta$. The two terms in the r.h.s can therefore be interpreted as two different sources of feedback, a first one controlled by the mean values $\langle m \rangle$, $\langle n \rangle$, and a second one controlled by the covariance $\sigma_{mn}$ between $\boldsymbol{m}$ and $\boldsymbol{n}$. Previous works analyzed the bifurcations in networks with a symmetric transfer function [49], or positive transfer function with non-zero $\langle m \rangle$ and $\langle n \rangle$ [49, 81]. The respective contributions of the two sources of feedback in networks with a positive transfer function have so far not been examined.

To extend previous studies, we therefore analyzed the bifurcations obtained by separately increasing each source of feedback in Eq (6) in networks with a positive transfer function. For $\sigma_{mn} = 0$, the feedback is generated only by the first term, and we controlled it by changing $\langle n \rangle$ while keeping $\langle m \rangle$ fixed. As the non-linearity in that term is given by $\langle \phi \rangle (\kappa)$, which is a positive sigmoid (Fig 6A, insert), increasing $\langle n \rangle$ beyond a critical value leads to a bifurcation to two *asymmetric* states with low and high values of $\kappa$ (Fig 6A). Since the mean $\langle m \rangle$ of the vector $\boldsymbol{m}$ is non-zero, these two values of $\kappa$ correspond to two states with a low and a high population-averaged firing rate (Fig 6C), as usually found when positive feedback is high [49, 62, 81–83].

In contrast, when $\langle m \rangle = \langle n \rangle = 0$ and $\sigma_{mn} \neq 0$, the recurrent feedback is generated only by the second term in Eq (6), for which the non-linearity is given by $\kappa \langle \phi'(0, \Delta) \rangle$. Independently of the precise form of $\phi$, $\kappa \langle \phi'(0, \Delta) \rangle$ as function of $\kappa$ is in general symmetric around zero (Fig 6D, insert). In consequence, increasing $\sigma_{mn}$ beyond a critical value leads to the emergence of two *symmetric* fixed points for $\kappa$ (Fig 6D), which correspond to two activity states with different patterns of activity (Fig 6E), but identical population-averaged firing rates (Fig 6F).

In summary, a mean-field analysis of rate networks with unit-rank connectivity predicts two qualitatively different types of bifurcations and bistable states depending on whether the connectivity vectors $\boldsymbol{m}$ and $\boldsymbol{n}$ have zero or non-zero mean. We therefore examined whether these two types of bifurcations appeared when increasing the overlap between $\boldsymbol{n}$ and $\boldsymbol{m}$ in spiking networks with unit-rank connectivity added on top of a random EI background. Increasing $\langle n \rangle$ with non-zero $\langle m \rangle$ and zero $\sigma_{mn}$ is in fact equivalent to increasing the mean excitation in the underlying EI connectivity [81]. In agreement with previous studies [7], we found that this could lead to the emergence of an asymetric bistability between a low and a high average activity state (Fig 6G–6I). Increasing $\sigma_{mn}$ in networks with $\langle m \rangle = \langle n \rangle = 0$ instead gives rise to a bifurcation to two symmetric activity patterns with equal population-averaged firing rates (Fig 6J–6L). The predictions of the mean-field analysis in low-rank rate networks were therefore directly verified in spiking networks, and allowed us to identify a novel bifurcation to two symmetric states of activity. As for transient dynamics, these findings held also outside of the asynchronous irregular regime, when the neurons were sparsely synchronized or fired regularly.

## Geometry of nonlinear autonomous activity in rank-two networks

Going beyond unit-rank connectivity, we next examined non-linear autonomous dynamics in network with a rank-two structure. As before, we first summarize the analyses of rate networks performed in previous studies, and then compare the dynamics in spiking networks with those predictions.

A rank-two connectivity structure is defined by two pairs of vectors $(\boldsymbol{m}^{(1)}, \boldsymbol{n}^{(1)})$ and $(\boldsymbol{m}^{(2)}, \boldsymbol{n}^{(2)})$:

$$J_{ij} = \frac{1}{N}(m_i^{(1)} n_j^{(1)} + m_i^{(2)} n_j^{(2)}). \tag{7}$$

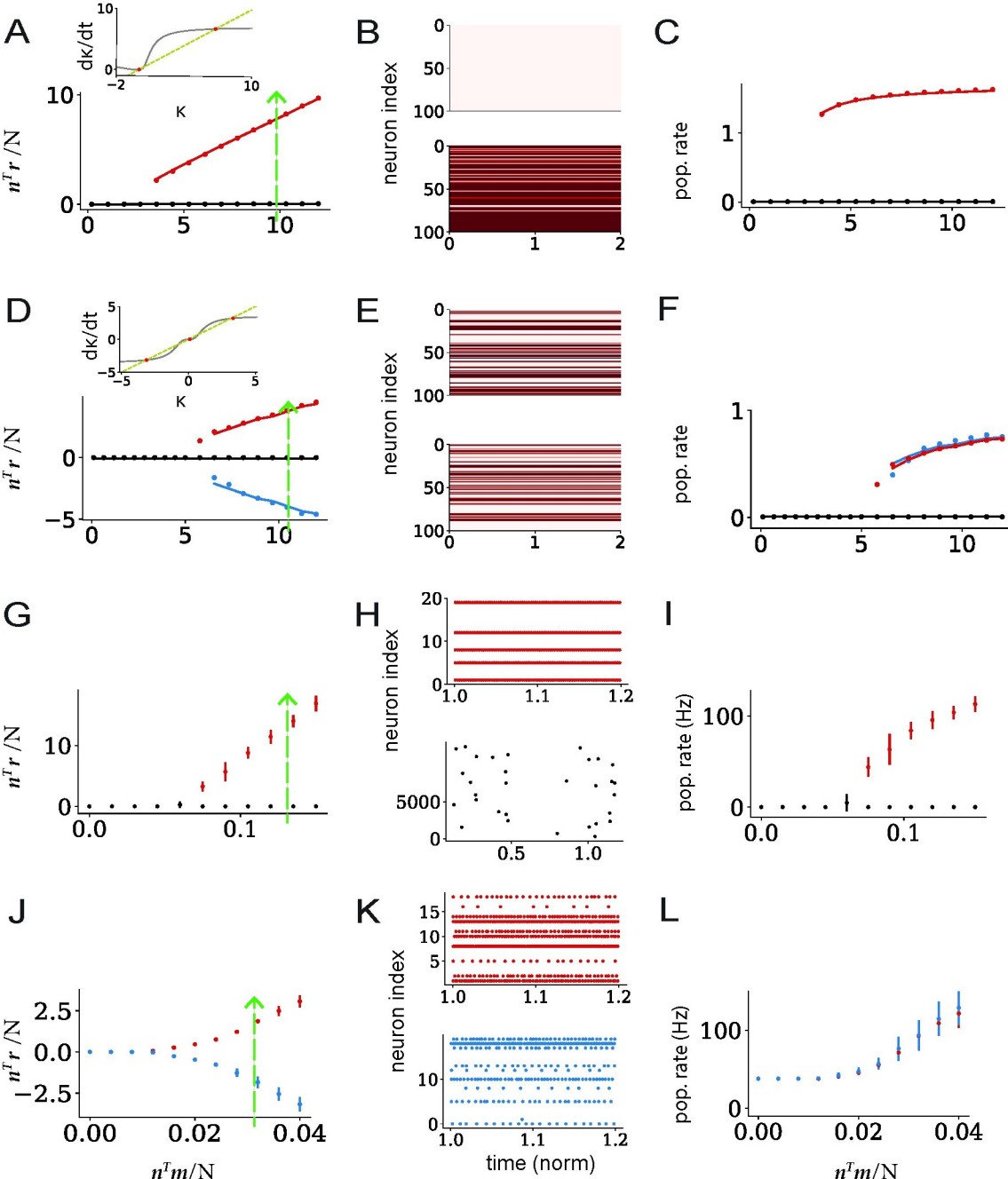

**Fig 6. Nonlinear autonomous activity in networks with unit-rank connectivity structure.** A-F: Rate networks. A-C: Connectivity vectors $\boldsymbol{m}$ and $\boldsymbol{n}$ with non-zero means $\langle m \rangle$, $\langle n \rangle$, and zero covariance $\sigma_{mn}$. A: Fixed points of the collective variable $\kappa$ as a function of the overlap $n^T m/N$, low (black) and high (red) activity state. Insert: RHS of the equation $d\kappa/dt$ (Eq (6)), $\kappa$ (yellow) and $\langle n \rangle \langle \phi \rangle (\kappa)$ (gray), shown for the overlap $n^T m/N = 10$. Fixed points (red dots) correspond to the intersections of $\kappa$ and $\langle n \rangle \phi(\kappa)$ which is a positive function. The bifurcation therefore leads to a low and a high state. B: Illustration of the single-unit firing rates in the two states when $n^T m/N = 10$ (dashed line in A, green) for 100 units. Top: low activity state. Bottom: high activity state. C: Population-averaged firing rate as a function of $n^T m/N$. D-F: same as A-C, for connectivity vectors $\boldsymbol{m}$ and $\boldsymbol{n}$ with zero means $\langle m \rangle$, $\langle n \rangle$, and non-zero covariance $\sigma_{mn}$. D: Fixed points of the collective variable $\kappa$ as a function of the overlap $n^T m/N$. Insert: RHS of the equation $d\kappa/dt$ (Eq (6)), $\kappa$ (yellow) and $\kappa \langle \phi' \rangle (\kappa)$ (gray), shown for the overlap $n^T m/N = 11.2$. Fixed points (red dots) correspond to the intersection of $\kappa$ and $\kappa \langle \phi' \rangle (\kappa)$, which is symmetric around the $y$ axis. The bifurcation therefore leads to two symmetric states (red and blue) on top of the low activity state. E: Illustration of the single-unit firing rates in the two symmetric states. F: Population-averaged firing rate as a function of $n^T m/N$. G-L: Simulations of the spiking network. G-I: connectivity vectors $\boldsymbol{m}$ and $\boldsymbol{n}$ with non-zero means $\langle m \rangle$, $\langle n \rangle$ and zero covariance $\sigma_{mn}$. G: bifurcation to low and high states as $\langle n \rangle$ is increased. H: raster plots of the spiking activity in the two states when $n^T m/N = 1.35$mV

(dashed line in J, green) for 20 neurons. Top: activity of 20 neurons in the high state. Bottom: activity of all (12500) neurons in the low state. The activity in the low state is highly sparse [7]. I: population-averaged firing rate in the two states. J-L: same as G-I connectivity vectors $\boldsymbol{m}$ and $\boldsymbol{n}$ with zero means $\langle m \rangle$, $\langle n \rangle$ and non-zero covariance $\sigma_{mn}$. J: bifurcation to two symmetric states as $\sigma_{mn}$ is increased. K: raster plots of the spiking activity in the two states when $n^T m/N = 32$mV (dashed line in J, green) for 20 neurons. L: population-averaged firing rate in the two states. Dots: simulations, lines: Monte Carlo integration predictions. Network parameters are shown in Tables 3 and 4.

In absence of external inputs, the activation dynamics $\boldsymbol{x}(t)$ are confined to the two-dimensional plane spanned by the vectors $\boldsymbol{m}^{(1)}$ and $\boldsymbol{m}^{(2)}$, so that, in analogy to Eq (3) the activation $x_i$ of unit $i$ can be expressed as:

$$x_i(t) = \kappa_1(t)\, m_i^{(1)} + \kappa_2(t)\, m_i^{(2)}. \tag{8}$$

Here $\kappa_1(t)$ and $\kappa_2(t)$ are two collective variables that describe the projection of $\boldsymbol{x}$ on the connectivity vectors $\boldsymbol{m}^{(1)}$ and $\boldsymbol{m}^{(2)}$.

Previous works [50, 51] have shown that in low-rank rate networks with Gaussian connectivity vectors, non-linear dynamics are fully determined by the eigenspectrum of the connectivity matrix. A rank-R matrix defined as in Eq (2) has in general R non-zero eigenvalues, that coincide with the eigenvalues of the $R \times R$ overlap matrix $\boldsymbol{P}^{ov}$ obtained from scalar products between pairs of connectivity patterns [50]:

$$P_{rs}^{(ov)} = \boldsymbol{n}^{(r)T}\boldsymbol{m}^{(s)}/N. \tag{9}$$

For rank-one networks, the overlap matrix reduces to a single parameter given in Eq (39), while for rank-two networks it is a $2 \times 2$ matrix. In the following, we focus on connectivity vectors with zero-mean entries, in which case for large N the overlap matrix converges to

$$\boldsymbol{P}^{ov} = \begin{pmatrix} \sigma_{n_1 m_1} & \sigma_{n_1 m_2} \\ \sigma_{n_2 m_1} & \sigma_{n_2 m_2} \end{pmatrix} \tag{10}$$

where $\sigma_{n_r m_s}$ is the covariance between the entries of vectors $\boldsymbol{n}^{(r)}$ and $\boldsymbol{m}^{(s)}$. A mean-field analysis then predicts that such networks have a fixed point at $(0, 0)$, the stability of which is determined by the eigenvalues of $\phi' \boldsymbol{P}^{ov}$ (Methods).

We specifically examined connectivity structures with two different forms of the overlap matrix, that lead to different configurations of eigenvalues and thereby generate qualitatively different types of nonlinear dynamics in rate networks [51].

We first consider rank-two networks with overlap matrices of the form:

$$\boldsymbol{P}^{(ov)} = \begin{pmatrix} \sigma & -\sigma_w \\ \sigma_w & \sigma \end{pmatrix}. \tag{11}$$

Such matrices have two complex conjugate eigenvalues $\sigma \pm i\sigma_w$. A mean-field analysis predicts spiral dynamics around the origin that decays to zero if $\phi'(0)\sigma < 1$, or generate limit cycle in the $\boldsymbol{m}^{(1)} - \boldsymbol{m}^{(2)}$ plane if $\phi'(0)\sigma > 1$. On the other hand, $\phi'(0)\sigma_w$ determines the frequency of these oscillations. Simulations of rate networks for $\phi'(0)\sigma > 1$ show that the firing rates of individual units oscillate strongly (Fig 7A, top), but out of phase, so that oscillations are not visible at the level of the population average (Fig 7A, bottom). Projecting $\boldsymbol{r}(t)$ on the $\boldsymbol{m}^{(1)} - \boldsymbol{m}^{(2)}$ plane however uncovers a clear limit cycle (Fig 7B) that corresponds to oscillations of $\kappa_1(t)$ and $\kappa_2(t)$ (Fig 7C).

To check whether qualitatively similar dynamics occur in spiking networks, we added a rank-two structure with complex eigenvalues on top of random excitatory-inhibitory

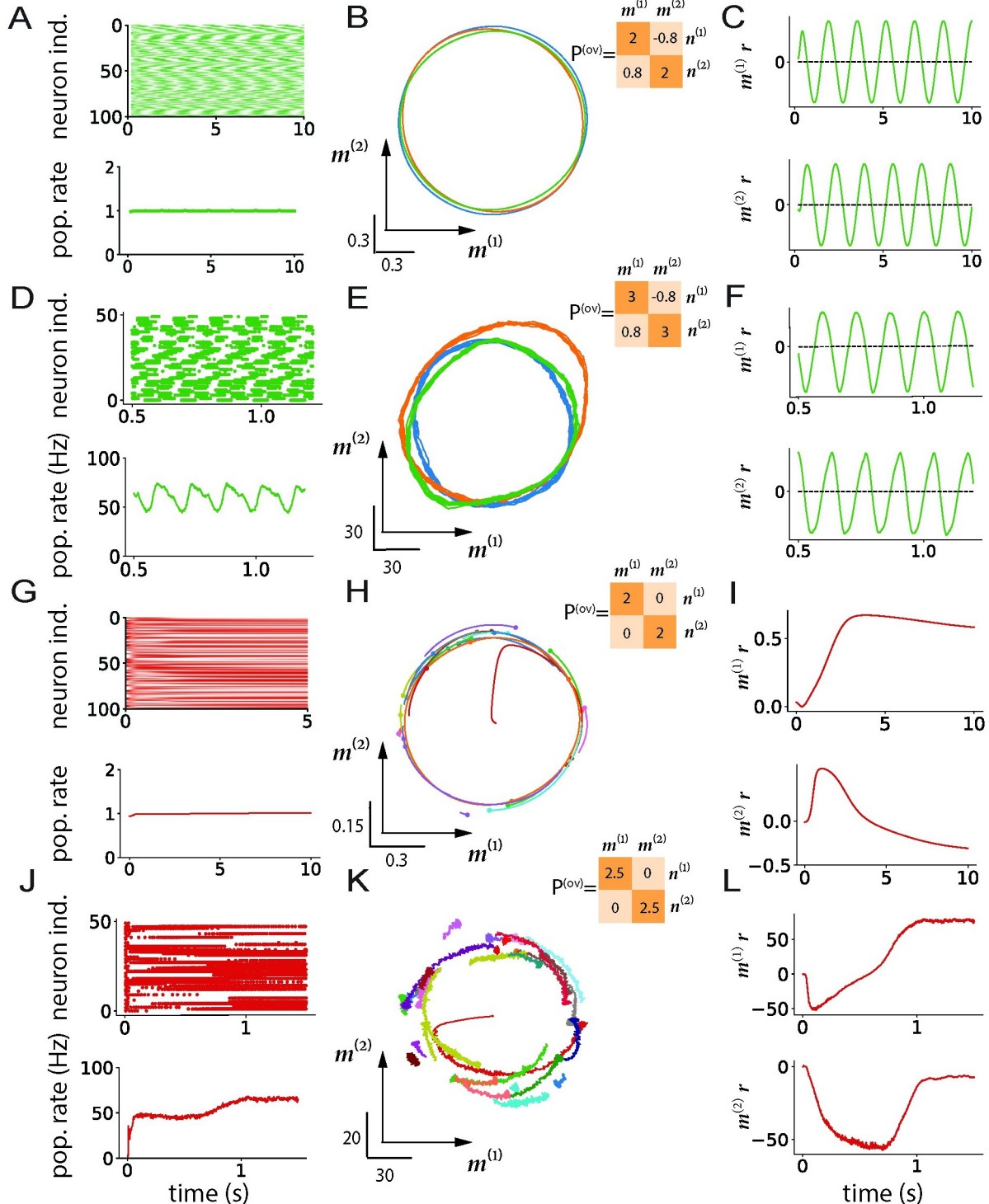

**Fig 7. Nonlinear dynamics in networks with rank-two structure.** A-F: Connectivity structure with two complex-conjugate eigenvalues. A-C: Rate networks. A: Top: Illustration of the single-unit firing rates for the first 100 neurons. Bottom: Population-averaged firing rate. B: Projections of the firing rates $r(t)$ on the $m^{(1)} - m^{(2)}$ plane. Insert: overlap matrix. C: Projections of the firing rates $r(t)$ on vectors $m^{(1)}$ and $m^{(2)}$ as a function of time. D-F: Analogous to (A-C), spiking network. D: Top: raster plots of the spiking activity for first 50 neurons. Bottom panel: population firing rate. E: Projections of the firing rates $r(t)$ on the $m^{(1)} - m^{(2)}$ plane. Insert: overlap matrix. F: Projections of the firing rates $r(t)$ on vectors $m^{(1)}$ and $m^{(2)}$ as a function of time. (G-I) Rate network dynamics for an overlap matrix that has two real, degenerate eigenvalues. G: Top panel: illustration of the single-unit firing rates for the first 100 neurons. Bottom panel: Population firing rate. H: Projections of the firing rates $r(t)$ on the $m^{(1)} - m^{(2)}$ plane. Insert: overlap matrix. I: Projections of the firing rate $r(t)$ on vectors $m^{(1)}$ and $m^{(2)}$

as a function of time. (J-L) same analysis as in (G-I) for a spiking model. J: Top panel: raster plots of the spiking activity for first 50 neurons. Bottom panel: population firing rate. K: Projections of the firing rates $r(t)$ on the $m^{(1)} - m^{(2)}$ plane. Insert: overlap matrix. L: Projections of the firing rate $r(t)$ on vectors $m^{(1)}$ and $m^{(2)}$ as a function of time. Different colors in the middle column (B,E,H,K) corresponds to network instances with different connectivity vectors but identical statistics. Network parameters are shown in Tables 5 and 6.

connectivity. As the two parts of the connectivity are independent, the spectrum of the full connectivity matrix consists of a continuous bulk corresponding to the random part and discrete outliers given by the eigenvalues of the rank-two structure [49, 84, 85]. For large values of $\sigma$, simulations of the resulting spiking network show that the firing rates of individual neurons oscillate strongly (Fig 7D, top), but out of phase with each other, so that oscillations on the population-averaged level are weak (Fig 7D, bottom). Projections of the population rate $r$ on the plane $m^{(1)} - m^{(2)}$ however identified clear limit cycles (Fig 7E and 7F).

We next turned to rank-two structure with overlap matrices of the form:

$$P^{ov} = \begin{pmatrix} \sigma & 0 \\ 0 & \sigma \end{pmatrix}. \tag{12}$$

The resulting connectivity matrices have two degenerate real eigenvalues $\sigma$, and mean-field analyses of rate networks have shown that in the limit $N \to \infty$, as $\sigma$ is increased this leads to a continuum of fixed points arranged on a ring in the $m^{(1)} - m^{(2)}$ plane [43, 49, 51]. In finite-size networks, sampling fluctuations of random connectivity vectors breaks the exact degeneracy, so that only a small number of points on the ring attractor remain actual stable fixed points while the rest form a slow manifold: dynamics quickly converge to the ring, after which they slowly evolve on it until reaching a fixed point (Fig 7H and 7I).

We verified that analogous dynamics emerge in spiking networks with a degenerate rank-two structure added on top of the random excitatory-inhibitory connectivity matrix. As in rate networks, dynamics quickly converge to a ring in the $m^{(1)} - m^{(2)}$ plane, after which they evolve along the ring towards stable fixed points (Fig 7K and 7L). Different instances of the rank-two structure generated with identical statistics lead to different fixed points that are all located on the same ring (Fig 7K). In spiking networks, the fluctuations in activity are stronger than in rate networks because of a combination of variability in spike times, random excitatory-inhibitory connectivity and fluctuations in low-rank connectivity, but the low-dimensional dynamics are qualitatively similar.

In summary, mean-field analyses of rate networks with low-rank connectivity allow us to identify analogous non-trivial dynamical regimes in networks of spiking neurons. These findings did not rely on the network being in the asynchronous irregular regime, as we observed similar dynamics when the neurons were sparsely synchronized (Fig 8C and 8E) or fired regularly (Fig 8D and 8F).

## Perceptual decision making task

Our results so far show that the geometry and firing regimes in networks of spiking neurons can be predicted from the statistics of low-rank connectivity by following the principles identified in rate networks. In a final step, here we illustrate how this finding can be exploited to directly implement computations in spiking networks. We consider the perceptual decision-making task [86] in which a network receives a noisy scalar stimulus along a random input vector $I$, and needs to report the sign of its temporal average along a random readout vector $w$.

Previous works have identified requirements on rank-one network to perform this task [52]. They showed that a unit-rank network was sufficient to implement the task, with

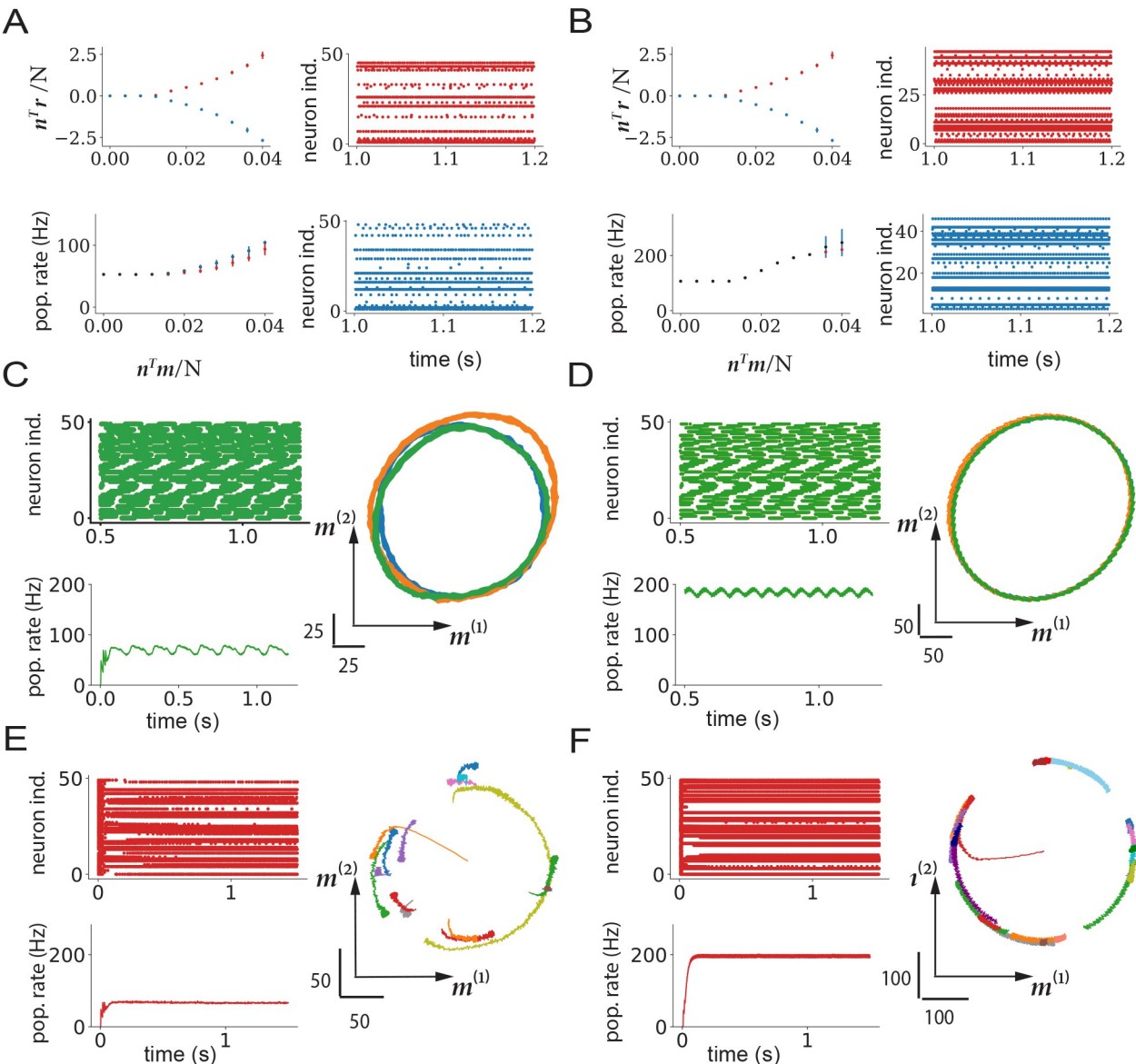

**Fig 8. Nonlinear dynamics in networks outside of the AI regime.** A,C,E: Network operating in the SI regime. A: nonlinear dynamics in rank-one network. Left, top: fixed points of the collective variable $\kappa$ as a function of the overlap $n^Tm/N$, for connectivity vectors $\boldsymbol{m}$ and $\boldsymbol{n}$ with zero means $\langle m \rangle$, $\langle n \rangle$, and non-zero covariance $\sigma_{mn}$. Left, bottom: population averaged firing rate as a function of $n^Tm/N$. Right: raster plots in the two symmetric states. C: non-linear dynamics in rank-two network, connectivity structure with two complex-conjugate eigenvalues. Left, top: raster plot of the spiking activity for first 50 neurons. Left, bottom: population firing rate. Right: projections of the firing rates $\boldsymbol{r}(t)$ on the $\boldsymbol{m}^{(1)} - \boldsymbol{m}^{(2)}$ plane. E: non-linear dynamics in rank-two network for an overlap matrix that has two real, degenerate eigenvalues. Left, top: raster plots of the spiking activity for first 50 neurons. Left, bottom: population firing rate. Right: projections of the firing rates $\boldsymbol{r}(t)$ on the $\boldsymbol{m}^{(1)} - \boldsymbol{m}^{(2)}$ plane. B,D,F: same as A,C,E for network without the background E-I connectivity.

connectivity statistics requiring a strong overlap $\sigma_{nI}$ to integrate inputs, and an overlap $\sigma_{mn} \approx 1$ to generate a long integration timescale via positive feedback. We built a spiking network based on an analogous connectivity configuration.

Fig 9 illustrates the dynamics in the network in response to two inputs with positive and negative means. The two inputs lead to different patterns of activity with opposite readout values (Fig 9A and 9B), but similar population averaged firing rates (Fig 9C). As expected from

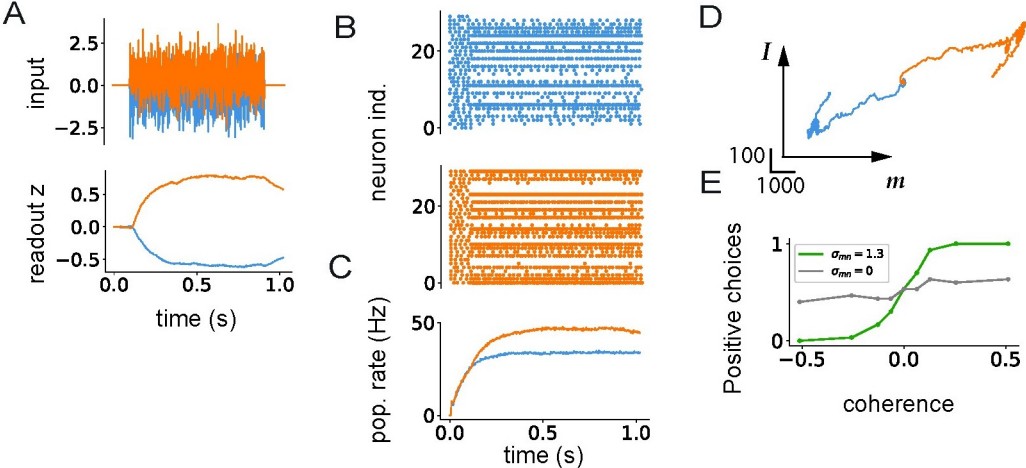

**Fig 9. Spiking network implementation of the perceptual decision-making task.** A: Top panel: two instances of the fluctuating input signal with a positive (orange) and a negative (blue) mean. Bottom panel: network readout of the activity generated by the two inputs. B: Raster plots for the first 50 neurons. C: Population firing rate. D: Dynamics projected onto the $I - m$ plane. E: Psychometric function showing the fraction of positive outputs at different values of the overlap $\sigma_{nm}$. Orange color corresponds to positive ($\overline{u} = 0.512$), while blue to negative mean-input ($\overline{u} = -0.512$). Parameters: $N = 12500$, $\sigma_u = 1$, $\sigma_{nI} = 0.26$, $\sigma_{mw} = 2.1$, $\sigma_{m^2} = 0.02$, $\tau_f = 100$ms.

the theory of low-rank networks, the dynamics evolve in a two-dimensional plane spanned by the input pattern $I$ and the output connectivity pattern $m$ (Fig 9D), as observed in experimental data [27]. The psychometric curve generated by the network strongly depends on the values of the connectivity overlaps (Fig 9E).

## Discussion

In this study, we set out to examine how far theoretical predictions for the relation between connectivity and dynamics in recurrent networks of rate units translate to networks of spiking neurons. To this end, we compared the population activity in rate networks with low-rank connectivity to networks of integrate-and-fire neurons in which a low-rank structure was added on top of random, sparse excitatory-inhibitory connectivity. Altogether, we found the geometry of low-pass filtered activity in spiking networks is largely identical to rate networks when the low-rank structure in connectivity is statistically identical. In particular, this allowed us to identify novel regimes of linear and non-linear dynamics in spiking networks, and construct networks that implement specific computations.

A widespread experimental observation across cortical areas is that sensory inputs lead to both increases and decreases of activity in individual neurons, so that different stimuli are often indistinguishable at the population-average level albeit they induce distinct patterns of responses [79, 87, 88]. Within the state-space picture, this implies that the responses take place primarily along directions orthogonal to the global axis [79], suggesting that behaviorally-relevant computations may rely on dynamics along these dimensions complementary to the population-average. So far, most studies on spiking networks have however focused on averaging spiking activity either over the whole network or over sub-populations. Here we instead show that, when a low-rank connectivity structure is included in the connectivity, spiking networks naturally lead to rich dynamics along dimensions orthogonal to the global axis. Our results therefore open the door to a closer match between spiking models and analyses of experimental data.

Our starting hypothesis was that spiking networks in the asynchronous irregular regime can be directly mapped onto rate networks with identical connectivity, by identifying each integrate-and-fire neuron with a rate unit. Here we tested a restricted version of this hypothesis by focusing exclusively on low-rank structure in the connectivity. We found that the population dynamics in spiking networks with a superposition of random and low-rank connectivity match well the predictions of rate networks with connectivity given by an identical low-rank part. To which extent these results extend to more general types of connectivity remains to be determined. A key feature of a low-rank connectivity structure is that it leads to discrete, isolated eigenvalues in the complex plane [47, 49, 80, 84, 85] (or singular values on the real line [89, 90], while purely random connectivity generates a continuously distributed bulk of eigenvalues [91, 92]. We expect that our findings hold as long as dynamics rely on discrete outliers in the the eigenspectrum (or singular value distribution), in which case the connectivity can be accurately approximated by a low-rank structure [81]. Networks performing specific computations typically rely on such outliers in the connectivity spectrum [21, 45, 93], so that our results may help explain in which case functional spiking networks can be directly built from trained rate networks [74–76].

A surprising result of our analyses is that rate networks predict well the activity in spiking networks even outside of the asynchronous irregular regime, i.e. when neurons spike regularly, or with some degree of synchrony. Our original hypothesis that asynchronous irregular activity is required appears to have been too restrictive. Indeed we found that our results hold as long as spike-trains are averaged over timescales longer than the synaptic or membrane time constants. When do spiking networks then qualitatively differ from their rate-based counterparts? Do spikes have a potential advantage over rate-based computations? One regime we have not explored here is ultra-sparse activity, in which each neuron emits only a handful of spikes in response to a stimulus. In this regime, information can be encoded in the precise timing of isolated spikes of individual neurons [94–96], and a comparison with state-space trajectories predicted by rate-based models may be less useful. The ultra-sparse firing regime provides a fruitful framework for energy-efficient neuromorphic computing [97], and suggests a potential computational role for spikes distinct from rate-based coding. An alternative possibility is that action potentials play mainly an implementational role, as a biological mechanism for transmitting information across long distances over myelinated axons, and therefore act as a discretization of a fundamentally continuous underlying signal. Ultimately, these computational and implementational interpretations of action potentials are not mutually exclusive, and it is possible that spikes may play different functional roles in different brain structures or species.

## Materials and methods

### Rate network model

We consider rate networks of $N$ units. Each unit is described by its activation $x_i(t)$, with dynamics evolving according to [98]:

$$\tau \dot{x}_i(t) = -x_i(t) + \sum_{j=1}^{N} P_{ij}\,\phi(x_j) + I_i u(t). \tag{13}$$

Here $u(t)$ is the input amplitude shared by all units, $I_i$ is the weight of the external input on unit $i$, and $\phi(x) = 1 + \tanh(x - x_{off})$ is the firing rate transfer function. The firing rate of unit $i$ is therefore $r_i = \phi(x_i)$.

The recurrent connectivity matrix $P$ is a rank $R$ matrix, represented as a sum of $R$ unit-rank terms, where the $r$-th term is given by the outer product of two vectors $m^{(r)}$, $n^{(r)}$:

$$P_{ij} = \frac{1}{N} \sum_{r=1}^{R} m_i^{(r)} n_j^{(r)}. \tag{14}$$

We refer to vectors $m^{(r)} = \{m_i^{(r)}\}_{i=1\ldots N}$, $n^{(r)} = \{n_i^{(r)}\}_{i=1\ldots N}$ as the right and left connectivity vectors, and to $I = \{I_i\}_{i=1\ldots N}$ as the input vector.

In this study, we focus on the case where the entries $m_i^{(r)}$, $n_i^{(r)}$, $I_i$ of connectivity and input vectors are generated independently for each unit from a Gaussian distribution with means $\langle m_r \rangle$, $\langle n_r \rangle$, $\langle I \rangle$, standard deviations $\sigma_{m_r}$, $\sigma_{n_r}$, $\sigma_I$ and covariances $\sigma_{xy}$ ($x, y \in \{n_r, m_r, I\}$).

To simulate network activity, Eq (13) was discretised using Euler's method with time step $dt$, for a total simulation time $t_{run}$. Network parameters are shown in Tables 1, 3 and 5.

## Spiking network model

We consider networks of $N$ leaky-integrate and fire neurons [7], where the membrane potential of neuron $i$ evolves according to:

$$\tau_m \frac{dV_i}{dt} = -V_i + \mu_0 + \sqrt{\tau_m} \sigma_0 \xi_i(t) + \mu_i^{rec}(t) + I_i\, u(t). \tag{15}$$

Here $\tau_m$ is the membrane time constant, $\mu_0$ a constant baseline input, $\xi_i(t)$ a white noise independent for each neuron, $\sigma_0$ the amplitude of the noise, $\mu_i^{rec}$ total recurrent input defined below, and $I_i$, $u(t)$ the weights and the amplitude of the external input.

An action potential, or "spike", is generated when the membrane potential crosses the threshold $V_{thr}$. The membrane potential is then reset to the value of $V_r$, and maintained at that value during a refractory period $t_{ref}$.

The total recurrent input to neuron $i$ is given by

$$\mu_i^{rec}(t) = \tau_m \sum_{j=1}^{N} J_{ij} \sum_k \delta(t - t_j^{(k)} - \tau_{del}) \tag{16}$$

where $J_{ij}$ is strength of the synaptic connection from neuron $j$ to neuron $i$, $t_j^{(k)}$ is the time of the $k^{th}$ spike of the presynaptic neuron $j$, $\tau_{del}$ is the synaptic delay and $\delta(t)$ is the delta function.

**Table 1. Parameters Fig 2.**

| Notation | Description |
| --- | --- |
| $N$ | 1000 |
| $\sigma_m$ | 1 |
| $\sigma_n$ | 1 |
| $\sigma_I$ | 1 (gray, blue), 0 (purple) |
| $\sigma_{nI}$ | 1 |
| $\langle I \rangle$ | 0 (gray, blue), 1 (purple) |
| $\tau$ | 100ms |
| $t_{run}$ | 5s |
| $dt$ | 1ms |
| $u(t)$ | input amplitude |

**Table 2. Parameters Figs 3 and 5.**

| Variable | Value | | | |
|---|---|---|---|---|
| $C$ | 1250 | | | |
| $\sigma_m$ | 2 | | | |
| $\sigma_n$ | 20mV | | | |
| $\sigma_{P^{(1)}}$ | 3.2$\mu$V | | | |
| $\tau_{ref}$ | 0.5ms | | | |
| $\tau_{del}$ | 1.5ms | | | |
| $\tau_f$ | $1-100$ms | | | |
| $t_{run}$ | 2s | | | |
| $\sigma_{nI}$ | 0.4mV$^2$ | | | |
| | Fig 3 | Fig 5A–5C | Fig 5D–5F | Fig 5G–5I |
| $J$ | 0.1mV | 0.1mV | 0.1mV | 0mV |
| $g$ | 5 | 6 | 5 | 5 |
| $\mu_0$ | 40mV | 80mV | 40mV | 30mV |
| $|I|$, global | 22.5mV | -/- | -/- | -/- |
| $|I|$, orthogonal | 125mV | -/- | /- | -/- |
| $|I|$, along | 125mV | 50mV | 50mV | 50mV |

The connectivity matrix $J$ consists of a sum of a full-rank excitatory-inhibitory part $J^{EI}$ and a rank-R matrix $P$:

$$J = J^{EI} + P. \qquad (17)$$

The matrix $P$ is identical to the rate model (Eq (14)), while $J^{EI}$ is a sparse, random excitatory-inhibitory matrix identical to [7]. Each neuron receives inputs from $C$ neurons, $C$ being much smaller of the total number of neurons $N$. The fraction of non-zero connections is $f_c = C/N = 0.1$, where 80% of incoming connections are excitatory and the rest are inhibitory. All non-zero excitatory synapses have the same strength $J$, while non-zero inhibitory synapses have the strength $-gJ$.

The network was simulated using the Euler method implemented in Brian2 package [99] with simulation step $dt$ and simulation time $t_{run}$.

**Table 3. Parameters Fig 6, rate model.**

| Variable | Value | |
|---|---|---|
| $\sigma_m$ | 2 | |
| $\sigma_n$ | 6 | |
| $\tau_m$ | 100ms | |
| $t_{run}$ | 20s | |
| $dt$ | 10ms | |
| $x_{off}$ | 2.9 | |
| | Panels A-C | Panels D-F |
| $\mu_m$ | 2 | 0 |
| $\mu_n$ | $0.1-6$ | 0 |
| $\sigma_{mn}$ | 0 | $0.1-10$ |
| $N_{nets}$ | 1 | 1 |
| $N_{tr}$ | 4 | 8 |

**Table 4. Parameters Fig 6, SNN model.**

| Variable | Value | |
|---|---|---|
| $\sigma_m$ | 2 | |
| $\sigma_n$ | 20mV | |
| $\sigma_{P^{(1)}}$ | 3.2$\mu$V | |
| $t_{run}$ | 1.2s | |
| $\tau_f$ | 100$ms$ | |
| | Panels G-I | Panels J-L |
| $C$ | 4000 | 1250 |
| $\mu_m$ | 0.01 | 0 |
| $\mu_n$ | $0 - 150$mV | 0 |
| $\sigma_{mn}$ | 0 | $0 - 40$mV |
| $J$ | 0.2mV | 0.1mV |
| $g$ | 4.8 | 5 |
| $\mu_0$ | 17.7mV | 40mV |
| $\tau_{ref}$ | 2ms | 0.5ms |
| $\tau_{del}$ | 2.5ms | 1.5ms |
| $N_{nets}$ | 5 | 7 |
| $N_{tr}$ | 3 | 2 |

Single-neuron firing rates were computed from spikes using an exponential filter with a time constant $\tau_f$. The instantaneous rate of $i$-th neuron at time $t$ is given by

$$\tau_f \frac{dr_i(t)}{dt} = -r_i(t) + \sum \delta(t - t_k)$$ 

(18)

where $\delta(t - t_k)$ is the delta function centered at $t_k$. In the case of multiple trials, rates $r_i$ are

**Table 5. Parameters Fig 7, rate model.**

| Variable | Value | |
|---|---|---|
| $\tau_m$ | 100ms | |
| $t_{run}$ | 10s | |
| $dt$ | 1ms | |
| | Panels A-C | Panels G-I |
| $\sigma_{m_1^2}$ | 1 | 1 |
| $\sigma_{n_1^2}$ | 7.24 | 7.24 |
| $\sigma_{m_2^2}$ | 1 | 1 |
| $\sigma_{n_2^2}$ | 3.63 | 3.63 |
| $\sigma_{m_1 n_1}$ | 2 | 2 |
| $\sigma_{m_1 n_2}$ | 0.8 | 0 |
| $\sigma_{m_2 n_1}$ | $-0.8$ | 0 |
| $\sigma_{m_2 n_2}$ | 2 | 2 |
| $N_{nets}$ | 3 | 15 |
| $N_{tr}$ | 5 | 3 |

averaged over trials:

$$\langle r_i \rangle = \frac{1}{N_{tr}} \sum_{k=1}^{N_{tr}} r_i(k) \tag{19}$$

If different values are used in a specific figure, these value are specified in a dedicated table (Tables 2, 4 and 6). The parameter notations for spiking models are summarized in Table 7. Parameters whose value do not change over different simulations/figures are given in Table 8.

## Geometry of responses to external inputs

To characterize the geometry of activity in the high-dimensional state space, we examined the projections of the firing rate trajectories $r(t) = \{r_i(t)\}_{i = 1\ldots N}$ on an arbitrary direction $w = \{w_i\}_{i = 1\ldots N}$, defined as:

$$\langle w^T r(t) \rangle = \frac{1}{N} \sum_{i=1}^{N} w_i r_i(t). \tag{20}$$

In particular, taking $w$ to be the global axis where $w_i = 1$ for all $i$, the projection gives the population-averaged firing rate:

$$\langle r \rangle(t) = \frac{1}{N} \sum_{i=1}^{N} r_i(t). \tag{21}$$

Based on previous works [49–52], below we summarize the predictions of low-rank rate models for the geometry of activity, and then describe a method for verifying these predictions using principal components analysis.

**Table 6. Parameters Fig 7, SNN model.**

| Variable | Value | |
|---|---|---|
| $C$ | 1250 | |
| $J$ | 0.1mV | |
| $g$ | 5 | |
| $t_{ref}$ | 0.5ms | |
| $\tau_{del}$ | 1.5ms | |
| $\tau_f$ | 20ms | |
| | Panels D-F | Panels J-L |
| $\sigma_{m_1^2}$ | 1 | 1 |
| $\sigma_{n_1^2}$ | 82.4mV | 26mV |
| $\sigma_{m_2^2}$ | 1 | 1 |
| $\sigma_{n_2^2}$ | 46mV | 26mV |
| $\sigma_{m_1 n_1}$ | 30mV | 25mV |
| $\sigma_{m_1 n_2}$ | 8mV | 0 |
| $\sigma_{m_2 n_1}$ | −8mV | 0 |
| $\sigma_{m_2 n_2}$ | 26mV | 25mV |
| $N_{nets}$ | 3 | 35 |
| $N_{tr}$ | 1 | 1 |
| $t_{run}$ | 1.2s | 3s |

**Table 7. List of notations for spiking network models.**

| Notation | Description |
|---|---|
| $N$ | number of neurons |
| $C$ | number of EI connections each neuron receives |
| $J$ | excitatory synaptic strength |
| $g$ | relative inhibition strength |
| $\tau_m$ | membrane time constant |
| $\mu_{rec}$ | total recurrent input |
| $\mu_0$ | baseline input |
| $\sigma_0$ | amplitude of the noise |
| $V_{thr}$ | threshold potential |
| $V_r$ | reset potential |
| $\tau_{ref}$ | refractory period |
| $\tau_{del}$ | synaptic delay |
| $\tau_f$ | filter time constant |
| $|I|$ | amplitude of the vector $I$ |
| $t_{run}$ | simulation run time |
| $\delta(t)$ | delta function |

**Rate networks.** In rate networks with a low-rank connectivity matrix, the dynamics of the activations $x(t) = \{x_i(t)\}_{i=1\ldots N}$ are explicitly confined to a low-dimensional subspace of state space [51, 52, 100], meaning that projections of $x(t)$ are non-zero only on vectors $w$ belonging to this subspace. Here we first reproduce the derivation of the geometry of the activations $x(t)$ [51, 52]. We then explore the implications for the geometry of firing rates $r(t)$ where $r_i(t) = \phi(x_i(t))$.

For low-rank connectivity, the dynamics of $x_i(t)$ are given by

$$\tau \dot{x}_i(t) = -x_i(t) + \frac{1}{N}\sum_{j=1}^{N}\sum_{k=1}^{R} m_i^{(k)} n_j^{(k)} \phi(x_j) + \sum_{s=1}^{N_{in}} I_i^{(s)} u_s(t), \qquad (22)$$

where for completeness we included $N_{in}$ scalar inputs $u_s(t)$ along input vectors $I^{(s)}$ with $s = 1\ldots N_{in}$.

We start by assuming that at time 0, the initial state $x(0)$ lies in the subspace spanned by $m^{(r)}$ and $I^{(s)}$, i.e. that $\langle w^T x(0)\rangle \neq 0$ if and only if $w$ is a linear combination of $m^{(r)}$ for $r = 1\ldots R$

**Table 8. Common parameters in Figs 3–9.**

| Variable | Value |
|---|---|
| $N$ | 12500 |
| $C$ | 1250 |
| $\tau_m$ | 20ms |
| $\tau_{ref}$ | 0.5ms |
| $dt$ | 1ms |
| $\sigma_0$ | 0.71mV |
| $V_{thr}$ | 20mV |
| $V_r$ | 10mV |
| $\tau_{del}$ | 1.5ms |

and $\mathbf{I}^{(s)}$ for $s = 1 \ldots N_{in}$. This assumption can be made without loss of generality. Indeed, if it is not full-filled, the initial state $\mathbf{x}(0)$ can be included as an additional input with $u_s(t) = \delta(t)$ and $\mathbf{I}^{(s)} = \mathbf{x}(0)$ in Eq (22).

It is then straightforward to show by induction from Eq (22) that for any $t$, $\langle w^T x(t) \rangle \neq 0$ if and only if $\mathbf{w}$ is a linear combination of $\mathbf{m}^{(r)}$ and $\mathbf{I}^{(s)}$. The activations $\mathbf{x}(t)$ therefore lie for any $t$ in the the subspace spanned by $\mathbf{m}^{(r)}$ and $\mathbf{I}^{(s)}$. Assuming for simplicity that these vectors form an orthogonal set, the activation of the $i$-th neuron $x_i$, can be written as:

$$x_i(t) = \sum_{r=1}^{R} \kappa_r(t)\, m_i^{(r)} + \sum_{s=1}^{N_{in}} v_s(t) I_i^{(s)}. \tag{23}$$

Here $\kappa_r$ and $v_s$ are scalar latent variables that correspond to the coordinates of $x(t)$ along the vectors $\mathbf{m}^{(r)}$ and $\mathbf{I}^{(s)}$, and can be computed by projecting $\mathbf{x}(t)$ on normalized directions $\mathbf{m}^{(r)}/||m^{(r)}||^2$ and $\mathbf{I}^{(s)}/||I^{(s)}||^2$:

$$\mathbf{x}^T \mathbf{m}^{(r)}/||m^{(r)}||^2 = \kappa_r \tag{24}$$

$$\mathbf{x}^T \cdot \mathbf{I}^{(s)}/||I^{(s)}||^2 = v_s. \tag{25}$$

Projecting Eq (22) on the vector $\mathbf{I}^{(s)}/||I^{(s)}||^2$, we then obtain

$$\tau \frac{dv_s}{dt} = -v_s + u_s(t), \tag{26}$$

while the projection on $\mathbf{m}^{(r)}/||m^{(r)}||^2$ gives

$$\tau \frac{d\kappa_r}{dt} = -\kappa_r + \frac{1}{N}\sum_{j=1}^{N} n_j^{(r)} \phi(x_j). \tag{27}$$

Inserting Eq (23) into Eq (27) then leads to the following dynamical system:

$$\tau \frac{d\kappa_r}{dt} = -\kappa_r + \kappa_r^{rec} \tag{28}$$

$$\kappa_r^{rec} = \frac{1}{N}\sum_{i=1}^{N} n_i^{(r)} \phi\left( \sum_{l=1}^{R} \kappa_l m_i^{(l)} + \sum_{s=1}^{N_{in}} v_s I_i^{(s)} \right). \tag{29}$$

To simplify notations, from here on we consider unit-rank networks with a single input ($R = 1$ and $N_{in} = 1$, we therefore drop the indices $r$ and $s$), where the entries of $\mathbf{m}, \mathbf{n}$ and $\mathbf{I}$ are generated from a joint Gaussian distribution with means $\langle m \rangle$, $\langle n \rangle$, $\langle I \rangle$, standard deviations $\sigma_m, \sigma_n, \sigma_I$ and covariances $\sigma_{xy}$ for $x, y \in \{m, n, I\}$. In the limit $N \to \infty$, the sum over $j$ in Eq (29) can then be replaced by an integral over the joint Gaussian distribution, which can be computed using Stein's Lemma for Gaussian integrals [50–52]. The dynamics for $\kappa$ then become (Methods):

$$\tau \frac{d\kappa}{dt} = -\kappa + \langle n \rangle \langle \phi(\mu, \Delta) \rangle + (\sigma_{nm}\kappa + \sigma_{nI}v)\langle \phi'(\mu, \Delta) \rangle \tag{30}$$

where the brackets denote the following Gaussian integral

$$\langle f(\mu, \Delta) \rangle = \int dx (2\pi)^{-\frac{1}{2}} \exp^{-x^2/2} f(\mu + \sqrt{\Delta}x)) \tag{31}$$

and

$$\mu = \langle I \rangle \nu + \langle m \rangle \kappa$$
$$\Delta = (\kappa \sigma_m)^2 + (\nu \sigma_I)^2. \tag{32}$$

We next turn to the geometry of firing rates $r(t)$ where $r_i(t) = \phi(x_i(t))$, and examine the projection of $r(t)$ on an arbitrary direction $w$ in the activity state space:

$$\langle w^T \phi(x) \rangle \quad = \frac{1}{N} \sum_{j=1}^{N} w_j \, \phi(x_j)$$
$$= \frac{1}{N} \sum_{j=1}^{N} w_j \phi(\kappa(t) m_i + \nu(t) I_i). \tag{33}$$

In linear networks (i.e. when $\phi(x) = x$), firing rates are equivalent to activations $x(t)$, and therefore their dynamics are confined to the subspace spanned by $m$ and $I$. The projection of $r(t)$ on any direction orthogonal to $m$ and $I$ is therefore zero. In particular, the projection on the global axis is non-zero only if $\langle I \rangle \neq 0$, or if $\langle m \rangle \neq 0$ and $\kappa \neq 0$.

Here we focus on non-linear networks, and directions $w$ with entries generated from a joint Gaussian distribution with entries of $m$ and $I$, specified by a mean $\langle w \rangle$, variance $\sigma_w^2$, and covariances $\sigma_{wm}$ and $\sigma_{wI}$. As for Eq (29), the r.h.s. of Eq (33) can be replaced by a Gaussian integral, and, using Stein's Lemma, be expressed as (see Methods):

$$\langle w^T r \rangle = \langle w \rangle \langle \phi(\mu, \Delta) \rangle + \sigma_{wm} \kappa(t) \langle \phi'(\mu, \Delta) \rangle + \sigma_{wI} \nu(t) \langle \phi'(\mu, \Delta) \rangle. \tag{34}$$

In Eq (34), the first term in the r.h.s. represents the population-averaged firing rate, i.e. the projection of $r(t)$ on the global axis. Indeed, taking $w_i = 1/N$ for $i = 1 \ldots N$, only the first term is non-zero. Moreover, Eqs (31) and (32) show that changes in the population averaged firing-rate $\langle \phi(\mu, \Delta) \rangle$ can be induced either through the mean input $\mu$ by non-zero means $\langle I \rangle$ and $\langle m \rangle$, or through the variance of the input $\Delta$ by non-zero variances $\sigma_I$ and $\sigma_m$ of the input and connectivity vectors.

The last two terms in Eq (34) respectively represent the projection of firing rates on the zero-mean parts of $m$ and $I$, i.e. changes in $r(t)$ along directions orthogonal to the global axis. Altogether, Eq (34) therefore predicts that the projection of the firing rate vector $r(t)$ is zero on any direction $w$ orthogonal to the global axis, $m$ and $I$. Interestingly, for Gaussian connectivity vectors considered here, the geometry of firing rate dynamics is therefore largely equivalent to linear networks (i.e. to the geometry of $x(t)$). The main difference is that in the non-linear case, the heterogeneity across neurons quantified by the input variance $\Delta$ can induce a non-zero component along the global axis even when $\langle I \rangle = 0$ and $\langle m \rangle = 0$.

These theoretical predictions are verified through simulations in Fig 2. The corresponding network parameters are given in Table 1.

**Principal component analysis.** In order to extract the low-dimensional subspace of the population activity from simulations, we performed dimensional reduction via a standard Principal Component Analysis (PCA). First, we construct the matrix $X$ in which every column corresponds to the time trace of firing rates $X[:, i] = r_i(t)$. The matrix $X$ is then normalized by subtracting the mean in every column. We compute the principal components (PCs) as the normalized eigenvectors of the correlation matrix $C = X^T X$, sorted in decreasing order of their eigenvalues $\lambda_i$. The activity matrix $X$ is then projected on the orthonormal basis generated by the PC vectors, yielding $X' = XE$ where $E$ is the $N \times N$ matrix with columns formed by PC components. The variance explained by each component is the

corresponding entry on the diagonal of the rotated correlation matrix $C' = X'^T X'$. For rate networks, we run PCA on individual trial, as we did not include noise in the dynamics. For spiking networks we run the PCA on firing rates averaged over trials $N_{tr}$ (see Tables 4 and 6).

### Geometry of nonlinear autonomous activity in unit-rank networks

**Rate network.** We now turn to the autonomous activity in unit-rank networks without external inputs. The autonomous dynamics of the collective variable $\kappa$ are described by Eqs (28) and (29) in which the external input is zero:

$$\frac{d\kappa}{dt} = -\kappa + \frac{1}{N}\sum_{i=1}^{N} n_i \phi(\kappa m_i). \tag{35}$$

Any fixed point $\kappa_0$ obeys:

$$\kappa_0 = F(\kappa_0). \tag{36}$$

where

$$F(\kappa) = \frac{1}{N}\sum_{i=1}^{N} n_i \phi(\kappa m_i), \tag{37}$$

The stability of $\kappa_0$ is determined by linearizing Eq (35), yielding:

$$\frac{d\kappa}{dt}\Big|_{\kappa=\kappa_0} = -1 + \frac{1}{N}\sum n_i m_i \phi'(\kappa_0 m_i). \tag{38}$$

The stability of $\kappa_0$ is therefore controlled by the overlap

$$\langle \boldsymbol{n}^T \phi' \boldsymbol{m} \rangle = \frac{1}{N}\sum_{i=1}^{N} n_i m_i \phi'(\kappa_0 m_i). \tag{39}$$

In the large $N$ limit, replacing the sum with a Gaussian integral and applying Stein's lemma, the r.h.s in Eq (37) can be further expressed as

$$F(\kappa) = \langle n \rangle \langle \phi(\mu, \Delta) \rangle + \sigma_{mn}\kappa\langle \phi'(\mu, \Delta)\rangle. \tag{40}$$

To examine the effects of the two terms in $F(\kappa)$, in the results we vary the overlap either by setting $\sigma_{mn} = 0$ and changing $\langle n \rangle$ or by setting $\langle m \rangle, \langle n \rangle = 0$ and changing $\sigma_{mn}$. To compute $F(\kappa)$, we approximate the Gaussian integrals $\langle \phi' \rangle, \langle \phi \rangle$ in Eq (40) using the Monte-Carlo method. Specifically, we choose an array of values for $\kappa$, and for each element compute the corresponding $F(\kappa)$ (Eq (37)) by averaging over 50 different realisation of vectors $\boldsymbol{m}$ and $\boldsymbol{n}$. We then determine the fixed point by solving for $\kappa = F(\kappa)$. The predicted population-averaged firing rate can then be computed as $\frac{1}{N}\sum_i \phi(\kappa m_i)$.

The corresponding results are shown in Fig 6. Network parameters are given in Table 3.

**Spiking network.** In Fig 6 the overlap is varied as in the rate network, either through $\langle n \rangle$ or the covariance $\sigma_{mn}$. We run the dynamics for $N_{nets}$ different network instances keeping the overlap $\boldsymbol{n}^T\boldsymbol{m}/N$ fixed, while resampling connectivity vectors $\boldsymbol{m}$ and $\boldsymbol{n}$ from Gaussians with mean $\langle m \rangle, \langle n \rangle$ and variance $\sigma_m, \sigma_n$ respectively. The dynamics for each network instance is run for $N_{tr}$ number of trials. In every trial, we resample the initial membrane potential $V(0)$ from a Gaussian distribution. At a fixed overlap, for each network configuration and at each

trial, the collective variable is computed as $\kappa_{curr} = \frac{1}{N}\sum_i r_i n_i$. To get the plot in Fig 6 we first set a threshold $\kappa_{tr}$ that is a boundary between zero state and the the high state (Fig 6G) or two symmetric states (Fig 6J). Then we average over all collective variables $\kappa_{curr}$ that have $|\kappa_{curr}| < \kappa_{thr}$ to compute low states. For those $|\kappa_{curr}| > \kappa_{thr}$, we average over all positive or over all negative $\kappa_{curr}$ values to get the high states or the two symmetric states. The parameters used for simulating spiking model in Fig 6 are presented in Table 4.

$$\tau\frac{d\kappa_1}{dt} = -\kappa_1 + \frac{1}{N}\sum_{i=1}^{N} n_i^{(1)}\phi(\kappa_1 m_i^{(1)} + \kappa_2 m_i^{(2)}) = G_1(\kappa_1, \kappa_2)$$

$$\tau\frac{d\kappa_2}{dt} = -\kappa_2 + \frac{1}{N}\sum_{i=1}^{N} n_i^{(2)}\phi(\kappa_1 m_i^{(1)} + \kappa_2 m_i^{(2)}) = G_2(\kappa_1, \kappa_2)$$

(41)

Assuming zero-mean Gaussian connectivity vectors, replacing sums by integrals in the $N \to \infty$ limit, and applying Stein's lemma leads to:

$$\frac{d\kappa_1}{dt} = -\kappa_1 + (\sigma_{n_1 m_1}\kappa_1 + \sigma_{n_1 m_2}\kappa_2)\langle\phi'(\mu, \Delta)\rangle$$

$$\frac{d\kappa_2}{dt} = -\kappa_2 + (\sigma_{n_2 m_1}\kappa_1 + \sigma_{n_2 m_2}\kappa_2)\langle\phi'(\mu, \Delta)\rangle$$

(42)

where

$$\mu = 0$$

$$\Delta = \sigma_{m_1^2}\kappa_1^2 + \sigma_{m_2^2}\kappa_2^2.$$

(43)

For zero-mean connectivity vectors, $(\kappa_1, \kappa_2) = (0, 0)$ is always a fixed point. A linear analysis shows that its stability is given by the eigenvalues of $\phi'(0)\mathbf{P}^{ov}$ [50, 51], where $\phi'(0)$ is the gain at zero, and $\mathbf{P}^{ov}$ the overlap matrix:

$$\mathbf{P}^{ov} = \begin{pmatrix} \sigma_{n_1 m_1} & \sigma_{n_1 m_2} \\ \sigma_{n_2 m_1} & \sigma_{n_2 m_2} \end{pmatrix}.$$

(44)

For Fig 7, we ran simulations for $N_{nets}$ network instances and $N_{tr}$ trials for each instance, and plot the projections without averaging over trials. The parameters used for simulating rate and spiking model in Fig 7 are presented in Tables 5 and 6.

## Perceptual decision-making task

We start from a network in AI regime as in Section Geometry of responses to external inputs, and add a unit rank structure on top of the random part.

In each trial, the model was run for $t_{run} = 1020$ms: a fixation epoch of duration $T_{fix} = 100$ms was followed by a simulation epoch of $T_{stim} = 800$ms, delay epoch of $T_{del} = 100$ms and a decision epoch $T_{dec} = 20$ms. The feed-forward input to neuron $i$ on trial $k$ was

$$I_i^{FF}(t) = I_i u^{(k)}(t)$$

(45)

where during the stimulation, $u^{(k)} = \overline{u}^{(k)}(t) + \psi^{(k)}(t)$, with $\psi^{(k)}(t)$ a zero-mean Gaussian white noise of standard deviation $\sigma_u = 1$. Connectivity vectors and the input vector were generated

from a Gaussian distribution with zero mean. The standard deviation of vector $\boldsymbol{m}$ was $\sigma_{m^2} = 0.02$, and the covariance between pairs of vectors $\sigma_{mn} = 0.016$, $\sigma_{nI} = 0.26$, $\sigma_{mw} = 2.1$. During the decision epoch, a single readout was evaluated by output of the network is defined by readout value:

$$z = \frac{1}{N} \sum_{j=1}^{N} w_i r_i \tag{46}$$

where $\boldsymbol{w}$ is a readout vector generated from a Gaussian with zero mean and standard deviation $\sigma_w = \sigma_{mw} / \sqrt{\sigma_{m^2} - \sigma_{mn}}$.

On trial $k$, if the mean of the readout if above zero, we label the output as 1, and as 0 otherwise. At every value of the overlap, psychometric curve is computed by plotting the fraction of trials that had an output 1. The network was run for 30 trials at each overlap.

## Mean-field theory and gaussian integrals

Using the mean-field theory, we derive in detail the projection in Eq (33) for the rank-one case, which can then be extended to higher ranks. Vectors $\boldsymbol{m}$, $\boldsymbol{I}$ and $\boldsymbol{w}$ are generated as

$$\boldsymbol{m} = \sigma_m \boldsymbol{X} \tag{47}$$

$$\boldsymbol{I} = \sigma_I \boldsymbol{Y}, \tag{48}$$

$$\boldsymbol{w} = \sigma_{mw}/\sigma_m \boldsymbol{X} + \sigma_{Iw}/\sigma_I \boldsymbol{Y} + \sqrt{\sigma_w^2 - (\sigma_{mw}^2/\sigma_m^2 + \sigma_{Iw}^2/\sigma_I^2)} \, \boldsymbol{Z} \tag{49}$$

where $\boldsymbol{X}$, $\boldsymbol{Y}$ and $\boldsymbol{Z}$ are independent vectors generated from a Gaussian distribution with zero mean and unit standard deviation, $\sigma_m$, $\sigma_I$, $\sigma_w$ standard deviations of vectors $\boldsymbol{m}$, $\boldsymbol{I}$, $\boldsymbol{w}$ and $\sigma_{mw}$, $\sigma_{Iw}$ overlaps of vectors $\boldsymbol{w}$ and $\boldsymbol{m}$, $\boldsymbol{I}$ respectively.

The dynamics in Eq (33) consist of a sum over the $N$ units in the network. In the limit of large networks with defined statistics, the sum over $N$ elements corresponds to the empirical average over the distribution of its elements. Therefore, we can replace the sum by an integral over the distribution $P(m, n, I)$.

$$
\begin{aligned}
\boldsymbol{w} \cdot \phi(x) &= \frac{1}{N} \sum_{j=1}^{N} w_j \phi(\kappa \, m_i + v_s I_i) = \\
&= \int dm \, dI \, dw \, P(m, I, w) \, w \, \phi(\kappa \, m + v_s I) \\
&= \int dX dY dZ P(X, Y, Z) w(X, Y, Z) \phi(\kappa \, m(X, Y, Z) + v_s I(X, Y, Z))
\end{aligned} \tag{50}
$$

We represented the integral in Eq (50) as a function of variables $X$, $Y$ and $Z$, which are independent, so that the joint distribution obeys $P(X, Y, Z) = P(X)P(Y)P(Z)$. Eq (50) then becomes:

$$\boldsymbol{w} \cdot \phi(x) =$$

$$= \iiint P(X)P(Y)P(Z)dXdYdZ\left(\frac{\sigma_{mw}}{\sigma_m}X + \frac{\sigma_{Iw}}{\sigma_I}Y + \sqrt{\sigma_w^2 - \frac{\sigma_{mw}^2}{\sigma_m^2} - \frac{\sigma_{Iw}^2}{\sigma_I^2}}\, Z\right) \cdot$$

$$\cdot \phi(\kappa_r\sigma_m X + \nu_s\sigma_I Y) =$$

$$= \frac{\sigma_{mw}}{\sigma_m}\int X\phi(\kappa_r\sigma_m X + \nu_s\sigma_I Y)P(X)dX\int P(Y)dY\int P(Z)d(Z)+$$

$$+ \frac{\sigma_{Iw}}{\sigma_I}\int Y\phi(\kappa_r\sigma_m X + \nu_s\sigma_I Y)P(Y)dY\int P(X)dX\int P(Z)d(Z)+$$

$$+ \sqrt{\sigma_w^2 - \frac{\sigma_{mw}^2}{\sigma_m^2} - \frac{\sigma_{Iw}^2}{\sigma_I^2}}\iint \phi(\kappa_r\sigma_m X + \nu_s\sigma_I Y)P(X)P(Y)dXdY\int ZP(Z)dZ =$$

$$= \sigma_{mw}\kappa_r(t)\langle\phi'(\mu_r, \Delta)\rangle + \sigma_{Iw}\nu_s(t)\langle\phi'(\mu_s, \Delta)\rangle$$

(51)

where $\mu_r = \kappa_r\sigma_m$, $\mu_s = \nu_s\sigma_I$ and $\Delta = (\kappa_r\sigma_m)^2 + (\nu_s\sigma_I)^2$. In the last line we use the Gaussian integral notation:

$$\langle f(\mu, \Delta)\rangle = \int dx(2\pi)^{-\frac{1}{2}}\exp^{-x^2/2}f(\mu + \sqrt{\Delta}x)$$

(52)

## Author Contributions

**Conceptualization:** Ljubica Cimeša, Srdjan Ostojic.

**Data curation:** Ljubica Cimeša.

**Formal analysis:** Ljubica Cimeša, Lazar Ciric.

**Funding acquisition:** Srdjan Ostojic.

**Investigation:** Ljubica Cimeša, Lazar Ciric.

**Methodology:** Ljubica Cimeša, Lazar Ciric, Srdjan Ostojic.

**Project administration:** Srdjan Ostojic.

**Resources:** Srdjan Ostojic.

**Software:** Ljubica Cimeša, Lazar Ciric.

**Supervision:** Srdjan Ostojic.

**Validation:** Ljubica Cimeša, Srdjan Ostojic.

**Visualization:** Ljubica Cimeša.

**Writing – original draft:** Ljubica Cimeša.

**Writing – review & editing:** Srdjan Ostojic.

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
