## [Decision Letter · Decision Letter 0]

21 Feb 2023

Dear Dr. Ostojic,

Thank you very much for submitting your manuscript "Geometry of population activity in spiking networks with low-rank structure" for consideration at PLOS Computational Biology.

As with all papers reviewed by the journal, your manuscript was reviewed by members of the editorial board and by several independent reviewers. In light of the reviews (below this email), we would like to invite the resubmission of a significantly-revised version that takes into account the reviewers' comments.

We cannot make any decision about publication until we have seen the revised manuscript and your response to the reviewers' comments. Your revised manuscript is also likely to be sent to reviewers for further evaluation.

Sincerely,

Peter E. Latham

Academic Editor

PLOS Computational Biology

Marieke van Vugt

Section Editor

PLOS Computational Biology

Reviewer's Responses to Questions

**Comments to the Authors:**

Reviewer #1: In this manuscript, the authors demonstrate empirically that several interesting properties of firing rate network models with low rank structure also hold in spiking network models. This is a clear, straightforward, and useful study that fills a gap in the literature. Pending some minor edits suggested below, I feel that this manuscript is appropriate for publication in PCB.

1) In the first couple of pages of Results, the models are described qualitatively in words, but equations for the models are not given. This is sometimes a reasonable choice of style for papers intended for an audience of biologists when the details of the model are secondary to the biological interpretation of the results. However, the audience for this manuscript is computational neuroscientists and the details of the model are central to the paper. Readers who are not familiar with this class of low-rank network models will need to flip back-and-forth between the Results and Methods to understand what is really going on. Whereas, most computational neuroscientists would only need a quick glance at a few equations up front in the Results section to understand the basic construction of the model.

Furthermore, even in the Methods, some details seem to be missing or at least they are difficult to infer from what is written.

I suggest that the authors put more details of the model in the Results section, particular in the first 2-3 pages where the model(s) are introduced. Some specific suggestions follow:

1a) Second sentence of the Results ("We quantify ..."):

Replace this sentence with Eq 12 from Methods (or the equivalent equation in vector form) and also add the equation r_i=phi(x_i) (OR r=phi(x) if using vector form). You can still explain what each variable represents, but it's much clearer to have the actual equations up front.

I personally prefer the vector form of the equations (instead of the component form used currently), but the authors can choose whichever form they prefer.

1b) In Methods, "The recurrent connectivity matrix J consists of a rank R structure P":

It is not clear what is meant by "consists of" here. Is J=P? If so, then why have two names for it? Or is J sparse with P the connection probability matrix? Or J=P+noise with J dense? Write out exactly how J is related to P. If J=P, then maybe it's better to have just one name for it?

Put this more precise definition of J and its relationship to P in the Results (replacing or complementing the discussion surrounding Eq 1, which also does not clarify the relationship between J and P).

1c) Are the J matrices generated in exactly the same way for the rate and spiking models? For spiking models, J=JEI+P where JEI is a sparse EI network (Eq 16). If J is not generated in exactly the same way for the rate models, then it is better to give the two J's different names and make their definitions and relationship more explicit and precise in the Results section.

I do not think that the equations defining the spiking model dynamics need to be given in Results. Those are fine just in the Methods, but the connectivity matrices should be defined.

2) Line 120 "Statistically identical low-rank connectivity structure"

Related to the last comment above, if J is generated in the same exact way then I think this is misleading. I see that "identical low-rank structure" could be interpreted to mean that only the low-rank part if statistically identical, but the phrase could also be interpreted to mean that the matrices themselves (which have a low-rank structure) are statistically identical. If the two J matrices are not statistically identical, please rephrase. For example, "Connectivity structures with statistically identical low-rank components" or something similar.

3) Line 229: "independently of the activity regime"

This is a little bit stronger than what you have shown. There are inevitably activity regimes that you have not tested. Instead, it would be better to write something like "in both the AI and SI regimes"

4) Lines 245-247:

Eq 4 here looks like an explicit equation for \\kappa, but it is an implicit equation. While the equation is correct, it might mislead some readers who interpret it as an explicit equation. Instead of writing "\\kappa is therefore given by" you can write "\\kappa therefore satisfies". And, after Eq 4, you can write "where x_i is the steady state value of x_i(t)" or something similar.

5) Lines 425-428 ("We expect that our findings hold..."):

I agree with the authors' expectation expressed here, but only under the additional assumption that the dynamics being studied should be related to the "discrete outliers" and not to the details of the eigenvalues inside the bulk itself. One can easily design examples where the low rank part does not capture the dynamics of interest. If the authors agree with this caveat, it might be worth mentioning. But I won't quibble if the authors disagree and choose not to incorporate this suggestion.

Reviewer #2: Chimes et al. present a comparison of the dynamics of randomly connected rate networks with low-rank connectivity components to those of excitatory-inhibitory integrate-and-fire networks with corresponding low-rank connectivity components. They find similar bifurcations in the spiking network as in the rate network. My major comments relate to discussing the theoretical basis for the analogy between the rate and spiking networks, how the regime of the underlying spiking network impacts the low-rank dynamics, and whether this impacts the variability of activity in the two networks.

1) The authors’ hypothesis that a rate network can be viewed as a high-dimensional mean field theory for a spiking network has some support in the derivation of rate networks as mean-field descriptions (i.e., neglecting fluctuations) of switching networks (e.g., Ginzburg & Sompolinsky 1994; Buice et al., 2006), and more recently in integrate-and-fire networks with stochastic spike emission (Ocker 2022).

A systematic mean field approximation of a deterministic integrate-and-fire network by a rate network is unknown (at least to me). The closest is the classic diffusion approximation. The spectral approach to the Fokker-Plnack dynamics in the diffusion approximation (Mattia & del Giudice 2006) might provide a more principled choice of rate network to match to the integrate-and-fire network. The sparse connectivity, weak connection strength (J=0.1 mV or 0.2 mV), and injected independent white noise suggest that the Fokker-Planck approach should be applicable. (While the Fokker-Planck equation resulting from the diffusion approximation is usually viewed as describing the distribution of membrane potentials across a population, it has also been used to describe the statistical density of single-neuron membrane potentials-for example in Trousdale et al., 2012.) In the absence of such a justification, why compare the rate network to an integrate-and-fire network instead of a compartmental network or some other model?

On a related note, the use of slow filters to map from the activity of a spiking network to a rate network also has some earlier precedent in the work of Pinto et al., 1996, who derived rate networks of this form under the assumption that synaptic interactions are slow relative to the membrane timescale.

2) Since the spiking and rate networks are quite different systems on their face, the existence of shared bifurcations is surprising. The classic heuristic derivations of rate models apply most closely to asynchronous irregular networks; do the dynamics of figures 5-7 depend on the regime of the spiking network?

3) Beyond showing that the LIF network has similar bifurcations of equilibria as the rate network, it would be useful to examine how their statistics differ. Do second-order statistics in the two networks (single-neuron and joint variability) exhibit the same trends in the two networks? And, do these depend on the underlying regime of the spiking network?

Minor comments:

The mean excitation and inhibition could be viewed as a low-rank component of the spiking network’s connectivity. Does the full connectivity matrix, J^{EI} + P in Eq. 16, still obey Dale’s law?

Line 140: What is the linear regime of the network?

Line 134: “i.e.” is usually followed by a comma (as is “e.g.”).

Line 270: Bistability between low and high states was also a feature of the early model of Ashby et al. 1962, as well as Amari (1971),

and a classic feature of the wilson-cowan equations (wilson & Cowan 1972).

Slow filters let rate network describe spiking network - similar to earlier works with slow synapses giving a rate description (e.g., harish & Hansel, ermentrout)

LIF networks with only low rank structure (?) scaling? All to all?

There were some references to “spiking noise” but the underlying spiking model is deterministic.

Reviewer #3: The authors add a low-rank matrix to the EI matrix of a spiking neural network and show that filtered spike trains follow similar low-dimensional dynamics as rate neurons connected by the same low-rank matrix. Various examples comparing rates networks with spiking networks are provided.

I have some concerns:

A) I would expect to see a comparison of the variance of the synaptic inputs into a neuron due to the EI matrix and that due to the low-rank matrix. What happens to the results if one increases or decreases the variance of the n component (sigma_n in Table 4). I suspect that currently the low-rank component dominates over the EI part of the weight matrix. The authors should titrate the effect of each component. Currently only removing the EI component has been studied in Fig 4G-I, but this is moot if the effect of the EI connectivity is already minor. Indeed, it appears that the hypothesis that neurons with Poisson / AI spiking are needed to represent rate neurons is not borne out, since removing the EI matrix yields similar low dimensional dynamics.

B) Even though the authors start with an EI matrix that satisfies Dale's law, it seems the final weight matrix would violate Dale's law since J=0.1mV and sigma_n=20mV. Can the authors maintain Dale's law and yet obtain their results? This will possibly require using a much smaller sigma_n and clipping weights. In either case, further study and discussion on this is needed for relevance to biology, especially since the authors start with an EI matrix.

**Have the authors made all data and (if applicable) computational code underlying the findings in their manuscript fully available?**

Reviewer #1: **No: **A github link is included, but the repository is empty.

Reviewer #2: None

Reviewer #3: None

PLOS authors have the option to publish the peer review history of their article (what does this mean?). If published, this will include your full peer review and any attached files.

Reviewer #1: **Yes: **Robert Rosenbaum

Reviewer #2: No

Reviewer #3: No
---

## [Decision Letter · Decision Letter 1]

27 Jun 2023

Dear Dr. Ostojic,

Easiest decision ever! Congratulations -- great paper!

Peter Latham, Associate Editor

---formal letter follows

We are pleased to inform you that your manuscript 'Geometry of population activity in spiking networks with low-rank structure' has been provisionally accepted for publication in PLOS Computational Biology.

Best regards,

Peter E. Latham

Academic Editor

PLOS Computational Biology

Marieke van Vugt

Section Editor

PLOS Computational Biology

Reviewer's Responses to Questions

**Comments to the Authors:**

Reviewer #1: The authors addressed all of my concerns. I have no further suggestions.

Reviewer #2: The authors have thoroughly addressed all my questions and comments.

Reviewer #3: The authors have addressed my concerns, and I am happy to recommend this work for publication.

**Have the authors made all data and (if applicable) computational code underlying the findings in their manuscript fully available?**

Reviewer #1: Yes

Reviewer #2: None

Reviewer #3: None

PLOS authors have the option to publish the peer review history of their article (what does this mean?). If published, this will include your full peer review and any attached files.

Reviewer #1: **Yes: **Robert Rosenbaum

Reviewer #2: No

Reviewer #3: No

---

## [Editor Report · Acceptance letter]

27 Jul 2023

PCOMPBIOL-D-22-01727R1 

Geometry of population activity in spiking networks with low-rank structure

Dear Dr Ostojic,

I am pleased to inform you that your manuscript has been formally accepted for publication in PLOS Computational Biology. Your manuscript is now with our production department and you will be notified of the publication date in due course.

With kind regards,

Zsofia Freund
